# LLM Probability Concentration: How Alignment Shrinks the Generative Horizon

## Abstract

Despite their impressive capabilities, aligned large language models (LLMs) often generate outputs that lack diversity. What drives this consistency in the generation? We investigate this phenomenon through the lens of probability concentration in the model's output distribution. To quantify this concentration, we introduce the *Branching Factor* (BF)–a token-invariant measure of the effective number of plausible next steps during generation. Our empirical analysis reveals two key findings: (1) BF often decreases as generation progresses, suggesting that LLMs become more predictable as they generate. (2) alignment tuning substantially sharpens the model's output distribution from the outset, reducing BF by nearly an order of magnitude (e.g., from 12 to 1.2) relative to base models. This stark reduction helps explain why aligned models often appear less sensitive to decoding strategies. Building on this insight, we find this consistency has surprising implications for complex reasoning. Aligned Chain-of-Thought (CoT) models (e.g., DeepSeek-distilled models), for instance, leverage this effect; by generating longer reasoning chains, they push generation into later, more deterministic (lower BF) stages, resulting in more stable outputs. We hypothesize that alignment tuning does not fundamentally change a model's behavior, but instead steers it toward stylistic tokens (e.g., "Sure") that unlock low-entropy trajectories already present in the base model. This view is supported by nudging experiments, which show prompting base models with such tokens can similarly reduce BF. Together, our findings establish BF as a powerful diagnostic for understanding and controlling LLM outputs - clarifying how alignment reduces variability, how CoT promotes stable generations, and how base models can be steered away from diversity.

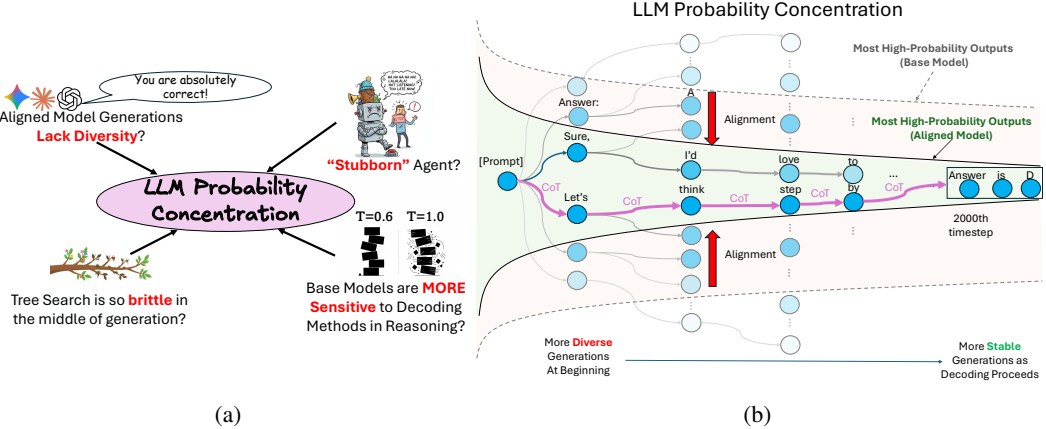

|  |  |
|:---:|:---:|
| (a) | (b) |

Figure 1: (a): LLM probability concentration connects and explains several disparate yet critical phenomena in aligned LLMs. (b): A conceptual illustration of how alignment and CoT influence the generation space of LLMs. While base models begin with high output diversity, alignment tuning sharply concentrates early probability mass, leading to more stable outputs. CoT extends this effect into later positions, flattening output sample variation and reducing sensitivity to decoding.

# 1 INTRODUCTION

While alignment tuning improves helpfulness and safety in large language models (LLMs), it often introduces a tradeoff: reduced output diversity (Padmakumar & He, 2024; Chakrabarty et al., 2024; Tian et al., 2024; Kirk et al., 2024; Lu et al., 2025) and increased determinism (Saparov & He, 2023; Song et al., 2024; Renze & Guven, 2024; Bigelow et al., 2024; West & Potts, 2025). Our own case study on MMLU (Hendrycks et al., 2021) confirms that aligned models exhibit reduced variance under Chain-of-Thought (CoT) prompting (Wei et al., 2022). These findings suggest a core phenomenon we call *LLM probability concentration* (Figure 1a) – i.e., a tendency to produce semantically and structurally similar outputs.

But how should we conceptualize this concentration? Autoregressive language generation is inherently a traversal through a *branching tree*: at each step, the model selects a token and expands the sequence along a specific branch. This structure, illustrated conceptually in Figure 1b, naturally defines a space of possible continuations, independent of specific token identities. To analyze how concentrated this space is during generation, we introduce the *Branching Factor* (BF) as a local, average-case measure of LLMs' output breadth, offering a microscopic lens on how local branching behavior gives rise to global output concentration.

To quantify BF, we leverage distribution perplexity (exponentiated entropy), an information-theoretic proxy for the effective size of a models output space.[1] Unlike standard perplexity evaluations that are computed over a dataset using teacher-forcing, our goal is to estimate the perplexity of the models *full output distribution* over continuations–i.e., the number of diverse, high-probability sequences it is likely to generate. However, directly computing this quantity is intractable due to the exponential number of possible outputs, especially for long generations. To address this, we leverage the Asymptotic Equipartition Property (AEP) (Shannon, 1948; Mudireddy et al., 2024), which implies that the average log-probability of sufficiently long samples approximates the length-averaged entropy of the underlying distribution. This lets us estimate BF directly from naturally sampled completions, without requiring teacher-forcing or exhaustive enumeration.

To see what variables control BF, we investigate how BF varies with output length, model size, prompt complexity, and training paradigms. Our findings reveal: ① BF typically declines over the course of generation, indicating that *model output becomes increasingly constrained, and thus more predictable, with each successive token.* ② Among various factors, alignment tuning (e.g., RLHF) exerts the strongest and most consistent impact, sharply compressing the branching factor by nearly an order of magnitude (e.g., $12 \rightarrow 1.2$). This pronounced narrowing *offers a quantitative basis for the reduced output variance and decoding insensitivity observed in aligned models*, highlighting a key behavioral divergence from their base counterparts. Our framework provides the probabilistic mechanism explaining *why* this occurs.

Is the BF reduction merely a statistical artifact, or does it indicate that the model is actively *honing in* on a narrower target space? To probe this, we conduct resampling experiments, requiring the model to choose an alternative to its top-ranked token at an intermediate generation step. We observe a sharp drop in accuracy, suggesting that aligned models not only concentrate probability mass but also commit to specific generative pathways early in the process. Building on these theoretical and empirical insights, we also hypothesize and empirically validate that aligned CoT models exhibit particularly low BF and reduced output variability under majority voting. This consistency arises because CoT encourages long reasoning chains, shifting key information to later tokens–precisely where BF tends to be lowest. As a result, different completions often converge to similar answers, *making CoT a natural stabilizer in generation.*

Why does alignment tuning exert such a dominant effect on BF? Inspired by the *superficial alignment hypothesis* (Zhou et al., 2024) and recent advances in tuning-free alignment (Lin et al., 2023; Fei et al., 2024), we hypothesize that base models already encode low-entropy conditional distributions, and that alignment tuning primarily steers generation toward certain stylistic tokens (e.g., "Sure"), thereby narrowing the conditional output space. To test this, we replicate nudging experiments (Fei et al., 2024) as a form of tuning-free alignment. We find that when conditioning base

---

[1]It is important to distinguish this from the standard NLP usage of "model perplexity," which measures how well a model fits a reference dataset (via exponentiated cross-entropy) (Jurafsky & Martin, 2025). Here, we measure the exponentiated entropy of the model's *own* output distribution.

models on prefixes typically produced by aligned models, BF drops more rapidly than when conditioning on self-generated prefixes. This supports our hypothesis that base models already contain low-entropy subspaces, which alignment surfaces rather than fundamentally reshaping.

In summary, by measuring LLM probability concentration via BF (§ 4), our contributions are:

① We find that aligned models possess a BF of around 1.2, nearly an order of magnitude lower than their base counterparts. This low BF helps explain reduced output diversity and randomness for aligned models. Also, the BF diminishes as generation progresses, reducing the influence of the decoding method further and suggesting LLMs become more predictable as they generate. (§ 5)

② Using this framework, we uncover an unexpected source of consistency in complex reasoning. We show that aligned CoT models, by generating extended reasoning chains, pushes generation into later, more deterministic (lower BF) regions. This indicates CoT stabilizes generations. (§ 6)

③ Perhaps most surprisingly, we find alignment surfaces low-entropy trajectories already latent in base models. Our evidence suggests when conditioning base models on low-probability prefixes typically produced by aligned models, BF drops more rapidly than under self-generated prefixes. This raises important questions about how to achieve alignment while preserving the rich generative capacity of base models. (§ 7)

## 2 BACKGROUND

**Autoregressive Language Models.** LLMs are typically trained to predict the next token and the probability of output $P(y_{1:N}|x;\theta)$ can be decomposed as: $P(y_{1:N}|x;\theta) = \Pi_{t=1}^{N} P(y_t|[x, y_{1:t-1}];\theta)$, where $y_{1:t-1}$ is the output up to position $t-1$, $\theta$ is the model parameter, and $x$ is the prompt. Each output sample is generated via token-by-token sampling, and the generation of multiple samples naturally forms a search tree (Yao et al., 2023; Hao et al., 2023; Wan et al., 2024). Modern LLMs would go through multiple training stages. In this paper, we would use *base models* to refer to the models trained without *alignment tuning* techniques (Touvron et al., 2023), including instruction tuning and Reinforcement Learning from Human Feedback (RLHF) (Ouyang et al., 2022; Bai et al., 2022) (e.g., "Llama-2-13B" (Touvron et al., 2023)) and *aligned models* to refer to models undergoing these additional fine-tuning stages (e.g., "Llama-2-13B-Chat").

**LLM Decoding and Entropy.** Though LLMs are trained with a large vocabulary size $|V|$, the desired tokens often concentrate on a much smaller set of tokens under distribution $P(y_t|x, y_{1:t-1}; \theta)$. Common decoding methods (Holtzman et al., 2020; Hewitt et al., 2022) utilize this observation and propose various heuristics to truncate vocabulary $V$ as $V_t$ at each step $t$. The next token is then sampled from the renormalized distribution $\tilde{P}(y_t|[x, y_{1:t-1}];\theta) = \mathbb{1}(y_t \in V_t)\frac{P(y_t|x, y_{1:t-1};\theta)}{\sum_{y_t \in V_t} P(y_t|x, y_{1:t-1};\theta)}$.

Since tokens are sampled from the truncated distribution $\tilde{P}$,[2] we use $\tilde{P}$ to compute the token-level entropy $\tilde{H}$ for a given prefix instance $y_{1:t-1}$:[3]

$$\tilde{H}(Y_t|[x, y_{1:t-1}];\theta) = -\sum_{y_t} \tilde{P}(y_t|[x, y_{1:t-1}];\theta) \log \tilde{P}(y_t|[x, y_{1:t-1}];\theta) \tag{1}$$

To generalize, we can compute the *expected conditional entropy* over the distribution of prefix sequences $Y_{1:t-1}$: $\tilde{H}(Y_t|[x, Y_{1:t-1}];\theta) = \mathbb{E}_{y_{1:t-1}} \tilde{H}(Y_t|[x, y_{1:t-1}];\theta)$. Conventionally, we use uppercase $Y$ to denote the *random variable* for an output and lowercase $y$ for its specific realization.

## 3 CASE STUDY: IS DECODING METHOD CRUCIAL FOR MODERN LLMS?

Many prevalent decoding methods were introduced before LLMs scaled to billions of parameters and underwent multiple training stages. Additionally, model developers adopt different decoding strategies when reporting LLM capabilities (Touvron et al., 2023; Dubey et al., 2024; Yang et al., 2024; Guo et al., 2025), raising questions about the significance of decoding choices for modern

---

[2]Our main experiments employ mild decoding settings ($T$=1.0, $p$=0.9). These settings approximate the full distribution, align with standard evaluation practices, and ensure coherent generation from base models. Stronger truncation settings are explicitly noted where applied.

[3]The common convention setting $0 \log 0 = 0$ for entropy computation is followed.

LLMs. To explore this, we benchmark various decoding methods on standard LLM reasoning tasks, extending prior work (Song et al., 2024; Renze & Guven, 2024; Shi et al., 2024a) to the latest models including DeepSeek-distlled models (Guo et al., 2025), which would generate long CoT before the final answer.[4] Specifically, we evaluate model performance on MMLU-STEM (Hendrycks et al., 2021) under CoT prompting across different temperatures ($T$= 0.6/1.0) in temperature sampling and truncation thresholds ($p$=0.9/1.0) in nucleus sampling (Holtzman et al., 2020). Further implementation details can be found in Appendix A.

| Models | Default ($T$=0.6, $p$=0.9) | $T$=0.6, $p$=1.0 | $T$=1.0, $p$=0.9 | Min ($T$=1.0, $p$=1.0) | $\frac{\text{Default}-\text{Min}}{\text{Default}}\%$ |
|---|---|---|---|---|---|
| Llama-3-70B-Instruct | 78.50 ($\pm$ 2.09) | 77.60 ($\pm$ 2.23) | 77.50 ($\pm$ 2.60) | 75.90 ($\pm$ 2.85) | 3.31 |
| Llama-3-70B | 78.00 ($\pm$ 3.52) | 74.00 ($\pm$ 3.80) | 72.00 ($\pm$ 4.38) | 63.50 ($\pm$ 5.02) | 18.59 |
| DeepSeek-R1-Distill-Llama-8B | 66.30 ($\pm$ 3.51) | 65.70 ($\pm$ 3.84) | 62.70 ($\pm$ 4.14) | 59.70 ($\pm$ 4.65) | 9.95 |
| Llama-3.1-8B-Instruct | 63.00 ($\pm$ 4.01) | 61.50 ($\pm$ 4.37) | 57.50 ($\pm$ 4.92) | 50.50 ($\pm$ 5.34) | 19.84 |
| Llama-3.1-8B | 54.00 ($\pm$ 4.61) | 53.50 ($\pm$ 4.92) | 47.00 ($\pm$ 5.21) | 37.00 ($\pm$ 5.48) | **31.48** |

Table 1: **Experiment Results across decoding methods on STEM subset of MMLU.** We follow the common practice of using 5-shot CoT prompting. $\frac{\text{Default}-\text{Min}}{\text{Default}}\%$ indicates the maximum relative performance drop when deviating from the default decoding configuration.

The results in Table 1 reveal that for aligned models, decoding configurations have a limited impact – typically around 10% (up to 20%) relative performance changes. Among Llama-3.1-8B models, DeepSeek-distilled Llama-8B (based on Llama-3.1-8B), which is trained to generate long CoT, exhibits the smallest relative performance changes. In contrast, base models exhibit greater sensitivity, with performance varying by up to 31%. Additionally, lowering the temperature ($T$) generally improves performance across all models more than adjusting truncation threshold ($p$), though excessive reduction (e.g., greedy decoding when $T$=0) may lead to repetition issues (Guo et al., 2025). Based on these observations and findings in existing literature, we propose the following hypotheses:

**Hypo 1** *Aligned models produce tokens with a more concentrated distribution than base models (Padmakumar & He, 2024; Bigelow et al., 2024; Lu et al., 2025; West & Potts, 2025).*

**Hypo 2** *Larger models have more concentrated distributions compared with smaller models (Ye et al., 2024; Xiong et al., 2024), though may varied by tasks (Lu et al., 2025; West & Potts, 2025).*

**Hypo 3** *As LLMs generate more tokens, its next-word prediction probability distribution becomes increasingly concentrated (Tian et al., 2024; Chakrabarty et al., 2024).*

Researchers often assess probability concentration using token-level metrics such as entropy or log-likelihood. However, these offer only a narrow lens on model behavior: they capture local properties but miss the global structure of the output space–how probability mass is distributed across plausible sequences. This motivates our proposal of the BF as a structural measure of generative breadth.

# 4 MEASURING BRANCHING FACTOR

**Probability Concentration on Effective Trees.** The generative process of language models can be viewed as moving down a branching tree, with each token choice selecting a path forward. While the full tree spans $O(|V|^N)$ sequences for vocabulary size $|V|$ and sequence length $N$, LLMs concentrate probability mass on a far smaller subset. This high-probability subset forms a complex, sparse "effective tree" $\mathcal{T}$. We propose to use the exponentiated entropy (perplexity) to quantify $|\mathcal{T}|$: $|\mathcal{T}| \stackrel{\text{def}}{=} \exp\left(H(Y_{1:N}|x;\theta)\right)$. This reflects the effective number of equally probable outcomes with the same total uncertainty (O'Connor, 2013). Analogously, it is like sampling from a fair $|\mathcal{T}|$-sided die, where entropy equals $-\sum \frac{1}{|\mathcal{T}|} \log \frac{1}{|\mathcal{T}|} = H(Y_{1:N}|x;\theta)$.

**Defining Branching Factor via Balanced Tree Model.** Since determining the exact structure of $\mathcal{T}$ is intractable, we *model* it as an equivalent balanced $B$-ary tree of the same depth $N$. A balanced tree with depth $N$ has $B^N$ leaf nodes. By equating this to the number of high-probability sequences $|\mathcal{T}|$ in the effective tree, we define the *Branching Factor* (BF) as $B \stackrel{\text{def}}{=} |\mathcal{T}|^{1/N}$. Thus, $B(x;\theta) =$

---

[4]For Llama-3 series models, in our prior study, we find there is only a minor performance difference between Llama-3 and Llama-3.x. We mainly use Llama-3 in this paper as it includes the most diverse model collection.

$\exp\left(\bar{H}\left(Y_{1:N}|x;\theta\right)\right)$ where $\bar{H}(Y_{1:N}|x;\theta) = \frac{1}{N}\tilde{H}\left(Y_{1:N}|x;\theta\right)$ is the length-averaged entropy up to position $N$. A larger $B(x;\theta)$ indicates a greater potential for diverse outputs.

**Variable Length and Monte-Carlo Estimator.** In practice, generation ends upon emitting an end-of-sequence (EOS) token, making output length $|y|$ a random variable. To avoid confounding effects and the need to fix an arbitrary generation length, we define the *length-marginalized* BF $\tilde{B}$ by aggregating over the output length distribution:

$$\tilde{B}(x;\theta) \stackrel{\text{def}}{=} \exp\left(\sum_N \underbrace{\tilde{P}(|y|=N|x;\theta)}_{\stackrel{\text{def}}{=} L(N|x;\theta)}\bar{H}(Y_{1:N}|x;\theta)\right) = \exp\left(\mathbb{E}_{N\sim L}[\bar{H}(Y_{1:N}|x;\theta)]\right)$$

$$= \exp\left(\mathbb{E}_{N\sim L}\left[\mathbb{E}_{Y_{1:N}}[\bar{H}(Y_{1:N}|x;\theta)|Y_N=\text{EOS}]\right]\right) \quad \text{(As } \bar{H}(Y_{1:N}|x;\theta) \text{ is constant for given } N)$$

$$= \exp\left(\mathbb{E}_{Y_{1:N}\sim\tilde{P}}[\bar{H}(Y_{1:N}|x;\theta)]\right)$$
$$(\text{As } \tilde{P}(y_{1:N}|x;\theta) = \tilde{P}(Y_{1:N}=y_{1:N}, Y_N=\text{EOS}|x;\theta) = L(N|x;\theta)\tilde{P}(y_{1:N}|x,Y_N=\text{EOS};\theta))$$

For small $N$, we can compute the term $\bar{H}(Y_{1:N}|x;\theta)$ exactly by exhaustive decoding. Lacking direct access to the output length distribution, we can use the above derivation to estimate the inner expectation via Monte-Carlo (MC) with $M$ independent samples $y^{(1)}, \ldots, y^{(M)}$:

$$\tilde{B}(x;\theta) \approx \exp\left(\frac{1}{M}\sum_{i=1}^{M}\bar{H}(Y_{1:|y^{(i)}|}|x;\theta)\right) \tag{2}$$

**Long Sequences and Estimator via Realized Entropy.** While effective for short outputs, the MC approach falters with longer sequences where the output space is exponentially large, causing systematic entropy underestimation (see Appendix C). Ideally, the Asymptotic Equipartition Property (AEP) (Shannon, 1948) would provide a solution, as it implies the length-averaged negative log-likelihood (NLL) of a sequence converges to the length-averaged total entropy. However, LLMs generally violate the stationarity and ergodicity assumptions required for AEP.

Fortunately, as noted by Mudireddy et al. (2024), a robust connection persists without these strict assumptions: the NLL converges to the *realized* entropy along a specific path, rather than the total entropy. Formally, let $h_{\text{realized}}(y_{1:N}) \stackrel{\text{def}}{=} \frac{1}{N}\sum_{t=1}^{N}H(Y_t|y_{<t};\theta)$ represent the length-averaged *realized entropy*. This relationship is characterized as follows:[5]

**Theorem 4.1 (Log-Likelihood Convergence for LLMs)** *Given* $0 < \epsilon < 1$*, as* $N \to \infty$*:*

$$P\left(\left|-\frac{1}{N}\log\tilde{P}(y_{1:N}|x;\theta) - h_{realized}(y_{1:N})\right| < \epsilon\right) \to 1 \tag{3}$$

Critically, while the length-averaged NLL of a *single* sequence converges to a random variable (the realized entropy) rather than the constant total entropy, the *expectation* of this realized entropy is exactly the total entropy: $\mathbb{E}_y[h_{\text{realized}}(y_{1:N})] = \bar{H}(Y_{1:N}|x;\theta)$. Combined with Equation (2), this motivates a hybrid estimator. For short sequences, we compute the exact entropy $\bar{H}$; for long sequences, we can validly substitute the computationally cheaper length-averaged NLL. Accordingly, we define $\tilde{B}$ using a length-dependent term $\mathcal{E}$:

$$\tilde{B}(x;\theta) \approx \exp\left(\frac{1}{M}\sum_{i=1}^{M}\mathcal{E}(y^{(i)})\right), \quad \mathcal{E}(y) = \begin{cases} \bar{H}(Y_{1:|y|}|x;\theta) & \text{if } |y| < L_\tau \\ -\frac{1}{|y|}\log\tilde{P}(y|x;\theta) & \text{otherwise} \end{cases} \tag{4}$$

This MC approach simultaneously handles length marginalization and entropy estimation.

As an empirical verification for Theorem 4.1, we plot NLL and realized entropy for sampled outputs of Llama-3-8B-Instruct over multiple datasets[6] in Figure 2. We observe that with increased output length, the difference between length-averaged realized entropy and NLL is reduced, and the standard deviation of length-averaged NLL also quickly reduces within the first $L_\tau = 10$ tokens. This indicates that despite non-ergodicity, the variance of NLL is sufficiently low for stable estimation.

Finally, for task-wise BF, we average all instance-wise BFs: $\tilde{B}(X;\theta) = \sum_x p(x)\tilde{B}(x;\theta)$.

---

[5]We provide a simplified proof in Appendix H with minor changes to the original proof of Mudireddy et al. (2024). Notably, our goal is only to show the approximation between length-averaged log-likelihood and entropy for a typical sequence, so we do not require stricter assumptions like ergodicity or stationarity.

[6]For dataset-specific details, we refer readers to Appendix B.

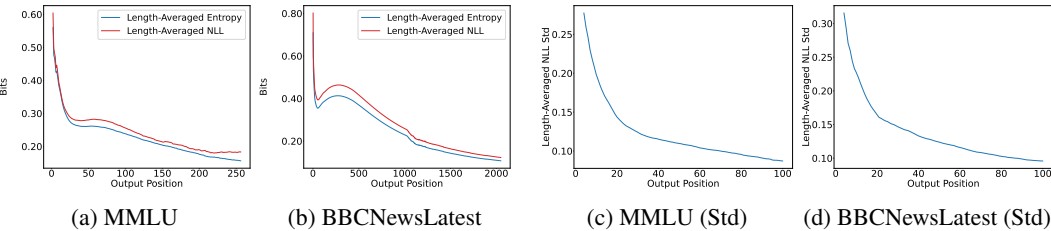

(a) MMLU      (b) BBCNewsLatest      (c) MMLU (Std)      (d) BBCNewsLatest (Std)

Figure 2: **Convergence of NLL and Entropy.** (**a, b**): The length-averaged NLL closely tracks the length-averaged Entropy. (**c, d**): The standard deviation of the length-averaged NLL diminishes rapidly with output length.

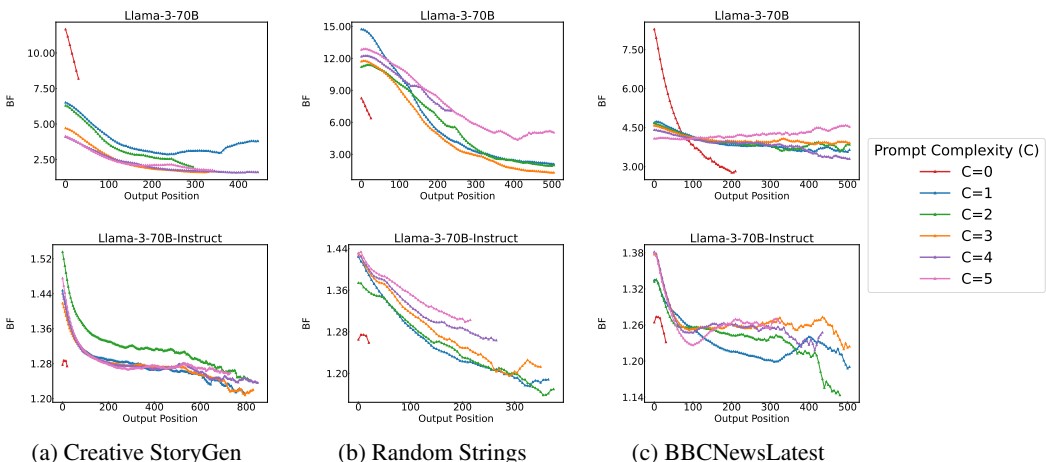

(a) Creative StoryGen      (b) Random Strings      (c) BBCNewsLatest

Figure 3: **Shrinking BF with output length over various tasks for Llama-3-70B and Llama-3-70B-Instruct.** For better visualization, we compute the exponential moving averaged values of BF with the smoothing factor set as 0.1.

## 5 BENCHMARKING AND ATTRIBUTING BRANCH FACTORS

**Models and Sampling.** We run experiments on models from Llama-2 (Touvron et al., 2023) and Llama-3 (Dubey et al., 2024) families as they are widely-used open-weight model families. For each model family, we include both base and aligned models to investigate how alignment tuning affects BF. We set $p$=0.9 and $T$=1.0 to sample outputs to conform with the setting for most datasets.

We set $M$=50 sequences to estimate BF, which yields a reliable estimation across datasets in prior studies. For aligned models, we apply the official chat templates to prompts. In addition, we carefully control the lengths of all inputs plus outputs to be within the context window of the models.

**Tasks.** We consider a variety of tasks covering common application scenarios of LLM generation, including reasoning and open-ended generation: MMLU (Hendrycks et al., 2021) (Reasoning), COGNAC (Chen et al., 2022) (Controlled Generation), BBCLATESTNEWS (Li et al., 2024b) (News Generation), and CREATIVE STORYGEN (Chakrabarty et al., 2024) (Creative Generation). To test subjective randomness bias (Bigelow et al., 2024), we also prepare a synthetic task RANDOM STRINGS where the prompt is generated via random characters. See Appendix B for dataset details.

**Impact Factors (IFs).** We consider modulating these factors that may impact BF computations: PROMPT COMPLEXITY ($C$), ALIGNMENT TUNING ($AT \in \{\text{Instruct}, \text{Base}\}$), MODEL SIZE ($S \in \{8\text{B}/13\text{B}, 70\text{B}\}$), and MODEL GENERATION ($G \in \{2, 3\}$). $C$ controls the informativeness of the input prompt $x$ (e.g., the number of banned words in Cognac, the number of in-context samples in MMLU). Intuitively, providing more information in $x$ should make the model more confident in its outputs, resulting in a lower BF. Dataset-specific setups for $C$ are detailed in Appendix B. $AT, S, G$ represent model-wise variations to explore how different configurations of $\theta$ affect $B(\text{X}; \theta)$.

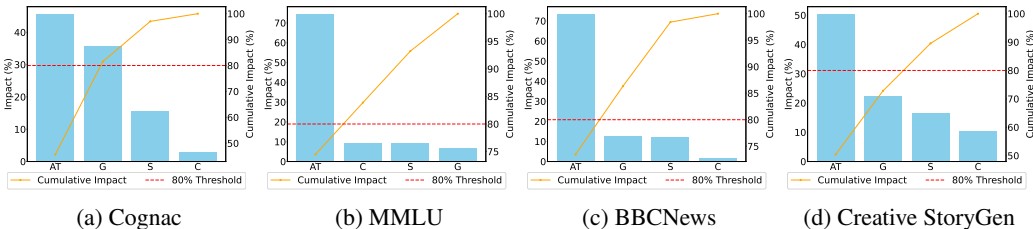

Figure 4: **Pareto Analysis of BF across various IFs.** $AT$ indicates whether the model is aligned. $C$ denotes the prompt complexity. $S$ refers to model size, and $G$ refers to model generation (Llama-2 vs. Llama-3).

## 5.1 BF DYNAMIC IN GENERATION PROCESS

Both BF and the output length $N$ are functions of the output Y, and BF computation relies on $N$. To avoid confounding effects, we first analyze how BF varies with $N$ before intervening IFs. In Figure 3, we demonstrate BF trajectories over different output positions by running Llama-3-70B and Llama-3-70B-Instruct on three representative tasks. Specifically, we compute BF over every five output tokens, conditioning on the prompt and all previously generated output tokens.[7] Our findings also generalizes to summarization, multilingual tasks, and Qwen family (Team, 2025) (Appendix G).

As we can see, first, **the average BF for the base model ( $\approx$ 12) is roughly ten times higher than the aligned model** ($\approx 1.2$).[8] Therefore, there are actually very few candidate next-token to be truncated in decoding for the aligned models. This explains why the decoding method would assert weaker effects for aligned models, as we see in § 3. Also, in most cases, **BF would often drop smoothly as more output tokens are generated**. Under the same task, when $C > 0$, different $C$ mainly controls the starting point and the rate of decreasing, while in the end, they would converge to roughly the same point. When almost zero knowledge is provided ($C = 0$), the output will end much earlier compared to $C > 0$ cases. These findings also provide support that the future token generation is gradually becoming predictable and the model may have a certain generation plan to follow, resonating with recent observation in interpretability (Pal et al., 2023; Wu et al., 2024; Li et al., 2024a) and inference acceleration (Cai et al., 2024; Welleck et al., 2024).

We further examine potential confounds such as prompt likelihood and data contamination in Appendix L, and find they do not fully account for the observed BF reductions.

## 5.2 PARETO ANALYSIS OF BF

We perform a Pareto analysis to identify the relative influence of all IFs of BF. For each factor $D_i$, we define the unnormalized *Impact* $\tilde{I}(D_i)$ as the average absolute pairwise difference in BF when varying $D_i$ while holding other dimensions constant:

$$\tilde{I}(D_i) = \frac{\sum_{d_i, d_j \in \text{Domain}(D_i), d_i \neq d_j} |\text{Avg}(\text{B}(\cdot|D_i = d_i)) - \text{Avg}(\text{B}(\cdot|D_i = d_j))|}{|\text{Domain}(D_i)| * |\text{Domain}(D_i) - 1|}. \tag{5}$$

Then we normalize it as $I(D_i) = \frac{\tilde{I}(D_i)}{\sum \tilde{I}(D_i)}$. The results, shown in Figure 4, indicate that **alignment tuning is the most influential factor affecting BF**. Across all tasks, it consistently crosses or approaches the 80% cumulative impact threshold either on its own or as the primary driver, surpassing model size, model generation, and prompt complexity by a large margin. For tasks with richer inputs–such as MMLU (with more in-context examples) and BBCLATESTNEWS (with more headlines)–prompt complexity $C$ and model size $S$ emerge as the next most impactful factors. In contrast, for open-ended tasks like Cognac and Story Generation, model generation $G$–particularly improvements from Llama-2 to Llama-3–plays a more dominant role. This shift likely reflects gains from the use of larger, more diverse datasets in training (Dubey et al., 2024).

Among these secondary factors, prompt complexity $C$ has a noteworthy effect: contrary to intuition, more context provided in the prompt does not always reduce BF but can in fact increase it, potentially

---

[7]See Appendix D for full results across all models and tasks.

[8]While the exact ratio varies by task and model, this "order of magnitude" difference is a frequent pattern in our results.

| Model | Maj@1 Std | Maj@3 Std | Maj@8 Std | Maj@16 Std | BF |
|---|---|---|---|---|---|
| DeepSeek-R1-Distill-Llama-70B | **14.34** | **8.29** | **4.99** | **3.21** | **1.23** |
| Llama-3-70B-Instruct | 16.37 | 11.40 | 7.50 | 5.12 | 1.28 |
| Llama-3-70B | 27.78 | 19.53 | 13.22 | 9.23 | 1.31 |
| DeepSeek-R1-Distill-Llama-8B | **27.10** | **20.91** | **13.93** | **9.14** | **1.23** |
| Llama-3.1-8B-Instruct | 31.54 | 24.64 | 17.30 | 12.90 | 1.31 |
| Llama-3.1-8B | 36.41 | 29.78 | 20.43 | 14.05 | 1.35 |

Table 2: **Majority Voting@K standard deviation on MMLU-STEM with** $200$ **samples.** We compute the standard deviation over 100 bootstrapping trials, each using 64 samples per instance. We set $T = 0.6, p = 0.9$ to match standard benchmarking settings, differing from $T = 1.0, p = 0.9$ setup in § 5. Lower temperature concentrates probability mass on fewer tokens, reducing BF and complicating direct comparisons. However, bootstrapping (100 runs) reveals minimal variability ($\approx 0.01$), confirming that the BF differences reported here remain significant. Consequently, BF remains a strong predictor of standard deviation.

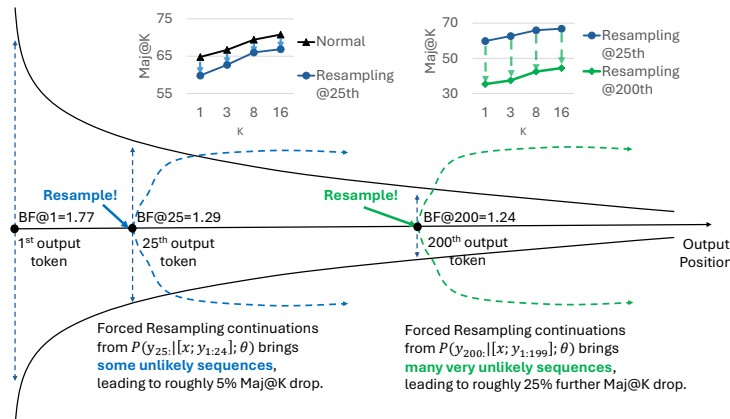

Figure 5: **Resampling from different output positions to assess the effect of interrupting BF reduction**. We resample new continuations at the 25th and 200th output token of DeepSeek-Distilled Llama-8B MMLU outputs. Results show substantial performance drops at both positions.

due to the cognitive burden of processing complex linguistic structures. A detailed case study and comprehensive task-wise BF results are presented in Appendices E and F.

# 6    Reduced BF Neglects Alternative Generation and Forking

Building on our findings that BF declines over the generation process (§ 5.1) and is lower in aligned models (§ 5.2), we derive a practical implication: aligned CoT models, by starting with low BF and delaying decisive tokens, shrink the output space more aggressively and produce fewer high-probability variants. To test this, we evaluate output variability on MMLU-STEM using 200 samples per model, measuring the standard deviation of Majority@K accuracy for $K = 1, 3, 8, 16$ under temperature $T = 0.6$ and truncation threshold $p = 0.9$. As shown in Table 2, among models with similar capacity, those with lower BF–especially the aligned CoT model–exhibit markedly lower variance. This confirms that BF is strongly correlated with sampling consistency.

But is this narrowing of BF merely a reflection of concentrated token probabilities, or does it reflect a deeper commitment to specific generative paths? To examine this, we conduct a resampling experiment: Procedurally, for a given position $t$: 1) Take the prefix $y_{<t}$ generated by the model. 2) Sample a new continuation $y'_{\geq t}$ from $P(Y_{\geq t}|[x, y_{<t}]; \theta)$. 3) Evaluate the full sequence $[y_{<t}, y'_{\geq t}]$ on the task. As shown in Figure 5, performance drops sharply when resampling occurs at a later, lower-BF position in the sequence. This suggests that aligned models are not just concentrating probability mass locally (reflects a "deeper commitment" to specific paths), but are actively locking into trajectories, making late-stage deviations more error-prone. In practice, this highlights a key application of BF: **parallel sampling should be applied early, while BF remains high**, to ensure meaningful diversity and avoid quality degradation.

# 7    How does Alignment Tuning Impact BF?

Why does alignment tuning exert such a pronounced effect on BF? Building on the superficial alignment hypothesis (Zhou et al., 2024) ("*Alignment tuning might simply teach base LLMs to select a subdistribution of data formats for interacting with users.*") and recent tuning-free alignment work (Lin et al., 2023; Fei et al., 2024; Lake et al., 2025), we hypothesize base models already encode low-entropy conditional distributions. In this view, alignment tuning doesn't reshape generation from scratch, but instead nudge the model toward *stylistic tokens* (e.g., "Sure"), thereby narrowing the conditional distribution.

To test this hypothesis, we reproduce the nudging experiments (Fei et al., 2024), over Just-Eval-Instruct (Lin et al., 2023) and MMLU datasets. We employ Llama-3-70B for drafting most outputs. However, when the base model's Top-1 probability is low, we apply nudging by switching to Llama-3-8B-Instruct to generate a single word. Using a smaller aligned model to nudge a larger base model (70B) isolates the steering effect of the stylistic prefix itself, independent of the nudging model's raw capability. BF was computed as in prior experiments. The results, shown in Figure 6,[9] indicate that after most nudging occurs early in the generation process – indicating the prefix generated by the nudging model is of low probability. These observations collectively support our hypothesis. Considering that nudging not only reduces BF but also improves aligned model performance on these tasks (Fei et al., 2024), our results highlight the dual effect of alignment training: reducing BF while preserving or even enhancing task performance.

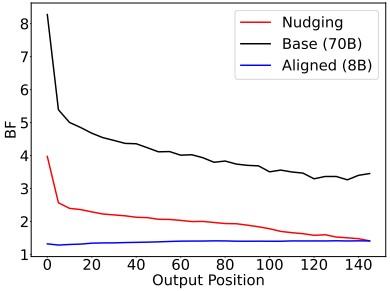

(a) Output BF Dynamics

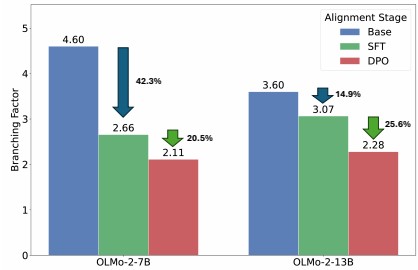

(b) Nudging Ratio Histogram

Figure 6: Nudging Experiments over Just-Eval-Instruct.

**Which Training Stage Reduces BF Most?** While our nudging analysis suggests alignment tuning narrows the distribution by nudging models toward stylistic tokens, it remains unclear which specific phase of the pipeline drives this reduction. Disentangling these effects is difficult because most model releases (e.g., Llama 3 (Dubey et al., 2024)) do not provide intermediate checkpoints, and modern post-training often employs iterative, interleaved schedules rather than discrete stages. However, the OLMo-2 suite (OLMo et al., 2024) releases intermediate checkpoints, enabling a preliminary stage-wise dissection. We measure BF changes across Supervised Fine-Tuning (SFT) and Direct Preference Optimization (DPO) (Rafailov et al., 2023) stages on the open-ended CREATIVE STORYGEN task. As shown in Figure 7, we observe distinct behaviors across model scales. For OLMo-2-7B, SFT is the primary driver of BF reduction (causing a 42.3% drop vs. 20.5% for DPO). In contrast, for OLMo-2-13B, the DPO stage contributes more significantly (25.6% drop vs. 14.9% for SFT). This divergence suggests that the impact of alignment tuning stages is not universal but sensitive to specific training recipes and model scales. Future work with fine-grained checkpoint trajectories is needed to fully map these dynamics.

Figure 7: **Stage-wise Contribution to BF Reduction.** We analyze OLMo-2 models on the Creative StoryGen task.

# 8 RELATED WORKS

**Uncertainty Quantification for LLM.** Uncertainty quantification (UQ) for LLMs has gained significant attention due to its importance in real-world applications, particularly in high-stakes domains (Desai & Durrett, 2020; Jiang et al., 2021; Wang et al., 2022; Kadavath et al., 2022; Xiong et al., 2024; Ye et al., 2024; Gupta et al., 2024). Existing methods typically address closed-domain

---

[9]We present results on Just-Eval-Instruct in short of space. MMLU results are included in Appendix I.

tasks such as classification and question-answering, where outputs are discrete and easier to assess. However, as Kuhn et al. (2023) note, these approaches often overlook challenges specific to open-ended generation, such as semantic equivalence across outputs. They introduce "semantic entropy" to quantify uncertainty in LLM output space by first clustering the sampled output and then quantifying uncertainty over cluster distribution. This method empirically works well in hallucination detection (Farquhar et al., 2024). In this paper, we focus on investigating the probability concentration phenomenon for LLMs. We introduce BF to quantify this concentration, which applies broadly across tasks without imposing strong assumptions on output categories.

**Reduced Diversity in Aligned Models.** Recent studies have consistently shown that alignment tuning reduces output diversity in language models (Perez et al., 2022; Padmakumar & He, 2024; Chakrabarty et al., 2024; Tian et al., 2024; Kirk et al., 2024; Lu et al., 2025; Lake et al., 2025; West & Potts, 2025). Mechanistically, Lake et al. (2025) observe that alignment suppresses diversity by aggregating information into longer, standardized responses, though they argue this preserves useful base model behaviors. Our work builds on this inquiry by connecting reduced diversity with related observations on diminished randomness and robustness in aligned models (Saparov & He, 2023; Song et al., 2024; Renze & Guven, 2024; Bigelow et al., 2024), and proposes a unifying explanation: increased probability concentration. Traditional diversity metrics such as n-gram lexical diversity (Li et al., 2016) are sensitive to vocabulary size and output length (Liu et al., 2022; Tevet & Berant, 2021; Guo et al., 2024; Kirk et al., 2024) and cannot work well with most recent long CoT models. In Appendix K, we demonstrate that lexical diversity poorly correlates with BF and fails to robustly measure generation concentration.

Our work also resonates with information density research in cognitive science and linguistic theories, and we present a short discussion in Appendix J.

# 9 DISCUSSION

**Practical Implications.** A key practical implication of our findings is that reduced BF neglects alternative generations and forking. Consequently, simply tweaking decoding parameters (e.g., temperature), is unlikely to restore diversity without severely degrading quality (Renze & Guven, 2024). Our work offers a clear explanation for why this occurs, particularly for methods like beam search. The resampling experiment in § 6 provides direct evidence that for low-BF models, off-path trajectories are not just less probable but often of lower quality. With little probability mass distributed among alternative paths, beam search has few viable options to explore, yielding diminishing returns. This suggests that efforts to mitigate diversity loss should target the training process itself – a more promising, albeit challenging, direction. Future work could involve curating more diverse alignment data or designing novel training objectives that balance instruction-following with distributional diversity (Wang et al., 2024; Kwon et al., 2024; Lanchantin et al., 2025; Chung et al., 2025). System-level interventions (e.g., model collaboration) also present a viable path forward (Fei et al., 2024; Lu et al., 2024; Venkatraman et al., 2025; Ismayilzada et al., 2025). While our paper's primary contribution is diagnostic, we believe this foundational understanding is a necessary prerequisite for developing such effective countermeasures.

**Societal Homogeneity Bias of Alignment Tuning.** Our work identifies a key dynamic in modern LLMs: alignment tuning significantly reduces the Branching Factor (BF), leading to more homogenized and predictable outputs. While this can be beneficial, it also carries potential negative societal impacts. In applications such as automated content generation, creative writing, or decision-support systems (Padmakumar & He, 2024; Sorensen et al., 2024; Wu et al., 2025a; Murthy et al., 2025; Rodemann et al., 2025; Ashkinaze et al., 2025; Lake et al., 2025), this reduction in diversity could inadvertently reinforce social biases, stifle creativity, and limit the exploration of novel ideas. Rodemann et al. (2025) further argue that empirical alignment, relying on limited and potentially biased human feedback, creates selectional bias that fails to capture the full spectrum of human values. We believe that formally understanding and quantifying the mechanisms of probability concentration, as we do in this paper, is a critical and necessary first step toward developing alignment techniques that mitigate these risks and foster models that are not only helpful and harmless but also diverse and robust.

REPRODUCIBILITY STATEMENT

To ensure the reproducibility of our findings, we will release all code used for our experiments upon publication. All datasets used in this paper are publicly available, with detailed descriptions, sources, and processing steps provided in Appendix B. Our experimental setup, including hyperparameters and model configurations, is detailed in § 5. The formal proof for the Asymptotic Equipartition Property for LLMs, central to our BF estimation, is provided in its entirety in Appendix H.

USAGE OF LARGE LANGUAGE MODELS

In this work, besides running LLMs in experiments, we use LLMs for the following purposes:

1. Aid or Polish Writing (Gemini 2.5 Pro, ChatGPT 4/5)
2. Literature Retrieval and Discovery (e.g., finding related work) (Gemini 2.5 Pro Deep Research, ChatGPT Deep Research)
3. Assisting Code Writing and Debugging (Claude 3.5 Sonnet)

We fully understand the responsibility of using LLMs in academic research. We carefully monitor any potential problems, such as plagiarism or scientific misconduct (e.g., fabrication of facts) when using LLMs. We make sure these problems do not occur in the paper.

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

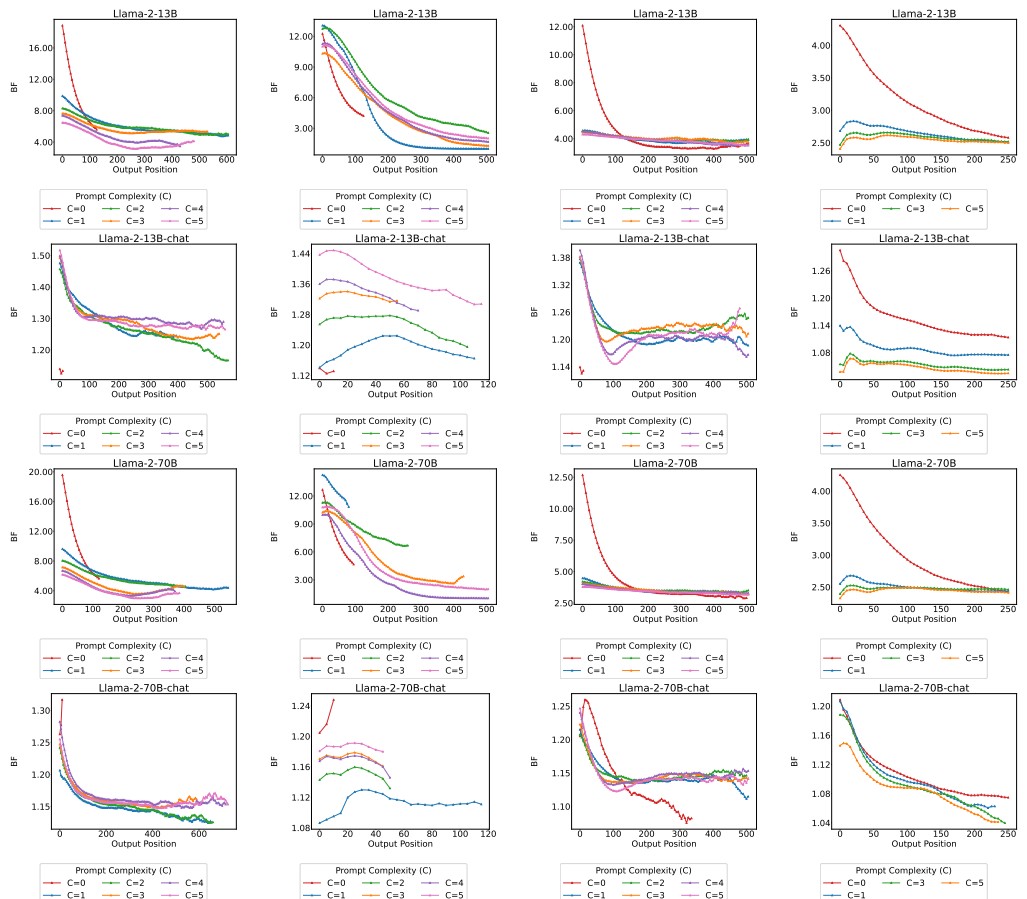

Figure 8: **BF Output Dynamic for Llama-2-families.** For better visualization, we compute the exponential moving averaged values of perplexity with the smoothing factor set as $0.1$.

## A CASE STUDY IMPLEMENTATION DETAILS

We use the scripts in Qwen-2.5-Math (Yang et al., 2024) for standard reasoning benchmarks.[10] We sample 200 examples from MMLU-STEM and compute the performance numbers under 64 trials and report the average performance.

## B DATASET-SPECIFIC PROCESSING

For all datasets we used in the paper, we carefully controlled whether the prompt length and expected output length would exceed the model's maximum length.

**MMLU** (Hendrycks et al., 2021) is a widely-used multiple-choice reasoning question. Unless otherwise explained, we use the full test set of MMLU to avoid potential contamination, following benchmarking settings reported in most LLM technical reports(Touvron et al., 2023; Dubey et al., 2024; Guo et al., 2025). We formulate prompt complexity $C$ as the number of in-context samples. For example, $C = 1$ means we only add one in-context sample. For prompting setup and postprocessing details, we follow the standard implementation in Qwen-2.5-Math (Yang et al., 2024).

**Cognac** (Chen et al., 2022) is a controlled generation task requiring language model *not* to generate specified banned words provided in the prompt. We use the WordNet subset (Miller, 1995) of Cognac as this is the only released setting in Cognac paper, where the topic is a root node and

---

[10]https://github.com/QwenLM/Qwen2.5-Math/tree/main

the constraint is defined as a subtree. We sampled 200 instances using the provided data generation codes in our experiments. To ensure most model generations ended properly in the decoding process, we relax the constraint of maximum decoded tokens $T$ from 60 to 512. We use the same prompt templates following their Github repo.[11]

**Creative Story Generation**   (Chakrabarty et al., 2024) provides the plots and story continuation from both machine and human. We adopt the provided 11 human-written story plots in the original dataset as the prompt. In this task, we set the maximum token $T = 1024$ to ensure the continued story written by LLM can have a proper ending. We formulate prompt complexity $C$ as providing $C \times 25$ words in the plot.

**Random Strings**   Similar to  Bigelow et al. (2024), we sample 200 random strings with length $L \sim U(256, 512)$ from the tokenizer vocabulary as the prompt. Prompt complexity $C$ is formulated by providing $C \times 15$ tokens in the prompt, ensuring each article contains at least 100 tokens.

**BBCLatestNews**    (Li et al., 2024b) is a news collection dataset aims at collecting news that is beyond the time cut for training LLMs. Unlike creative story plots, news articles are typically more structured and organized, although headlines can still be surprising. We select news articles from January to July 2024 to minimize data contamination, as the Llama models have a knowledge cut-off in late 2023. We formulate prompt complexity $C$ as providing $C \times 15$ words in the prompt.

## C    ENTROPY UNDERESTIMATION VIA MONTE CARLO SAMPLING

To demonstrate the limitations of Monte Carlo (MC) sampling for entropy estimation in long sequences, we conducted an empirical study. We prompted Llama-3-8B-Instruct with 5-shot CoT examples from the MMLU dataset. We then estimated the entropy of its generated responses using a varying number of MC samples: $M \in \{4, 8, 16, 32, 64\}$.

As illustrated in Figure 10, the estimated entropy consistently increases with the number of samples. This trend confirms that MC estimation with a small sample size systematically **underestimates** the true entropy because it fails to capture the long tail of the full probability distribution. While increasing the sample count mitigates this bias, it does so at a significant computational cost. In contrast, AEP Theorem 4.1 allows us to use the negative log-likelihood (NLL) of a single typical sequence for a more efficient and accurate estimation.

## D    FULL BF OUTPUT DYNAMICS INVESTIGATION

Here we present full task-wise and model-wise BF output dynamic for Llama-2 in Figure 8 and Llama-3 in Figure 9. We can observe the trends as in § 5.1: ① **The average BF for the base model ( $\approx 12$ ) is roughly ten times higher than the aligned model ($\approx 1.2$). ② BF would often drop smoothly as more output tokens are generated**.

## E    CURIOUS CASE OF PROMPT COMPLEXITY

Intuitively, greater prompt specificity (larger $C$) reduces BF by narrowing the models output space through more informative context. However, our experimental results reveal task-varied effects. As illustrated in Figure 11 for the Cognac task, greater prompt complexity can *increase* BF–potentially due to the cognitive burden of processing negation or complex linguistic structures. In contrast, for tasks like News Generation, higher $C$ generally leads to lower BF, consistent with the expected narrowing of output diversity. Comprehensive task-wise BF results are provided in Appendix F.

## F    FULL TASK-WISE BF EVALUATION ON DIFFERENT PROMPT COMPLEXITY

The full task-wise BF evaluation results over different prompt complexity can be found in Figure 12. Here we can see that prompt complexity modulates BF in highly non-consistent ways across models

---

[11]https://github.com/princeton-nlp/Cognac/tree/main

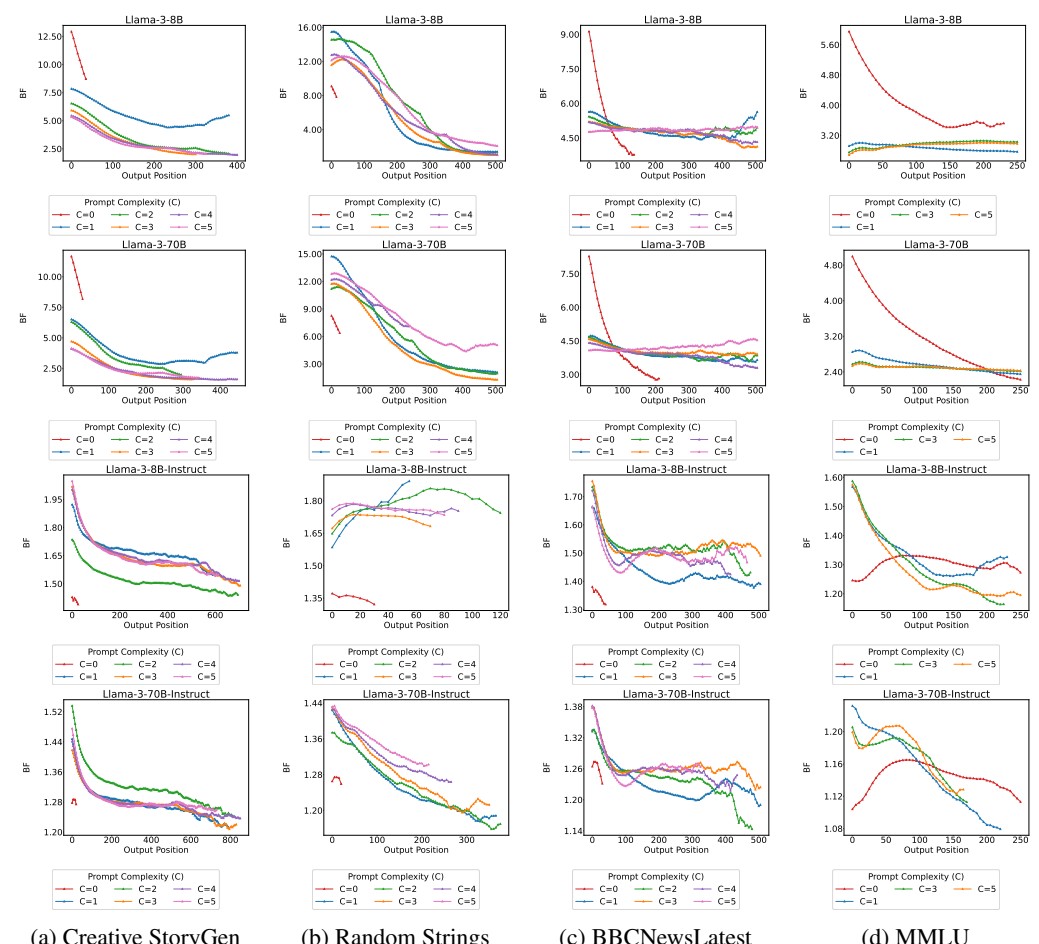

(a) Creative StoryGen  (b) Random Strings  (c) BBCNewsLatest  (d) MMLU

Figure 9: **BF Output Dynamic for Llama-3-families.** For better visualization, we compute the exponential moving averaged values of perplexity with the smoothing factor set as $0.1$.

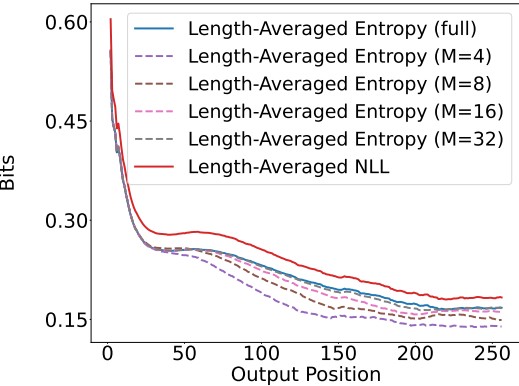

Figure 10: **Monte Carlo (MC) sampling systematically underestimates entropy.** The plot shows that the estimated entropy of sequences from Llama-3-8B-Instruct increases with the number of MC samples ($M$). A small sample size fails to cover the vast output space, leading to an underestimation of the true entropy. This bias is difficult to eliminate without incurring substantial computational costs.

and tasks, and there are no clear monotonic patterns, contradicting the intuition that with more context given, the model should have more confidence in what to generate.

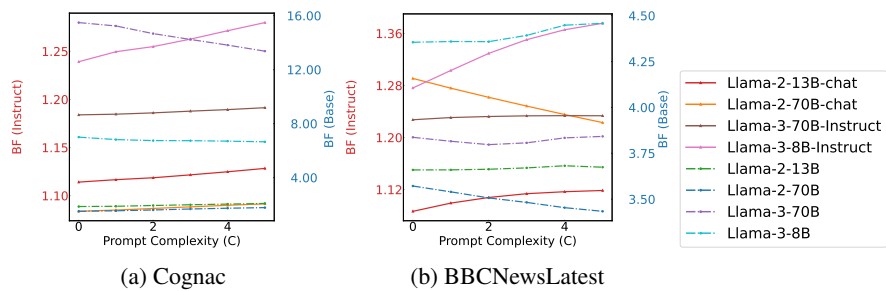

(a) Cognac       (b) BBCNewsLatest

Figure 11: **Task-varied influence of prompt complexity** $C$ **on BF.** On Cognac, we see BF increases with increased $C$, while on BBCNewsLatest, increasing $C$ can lead to reduced BF.

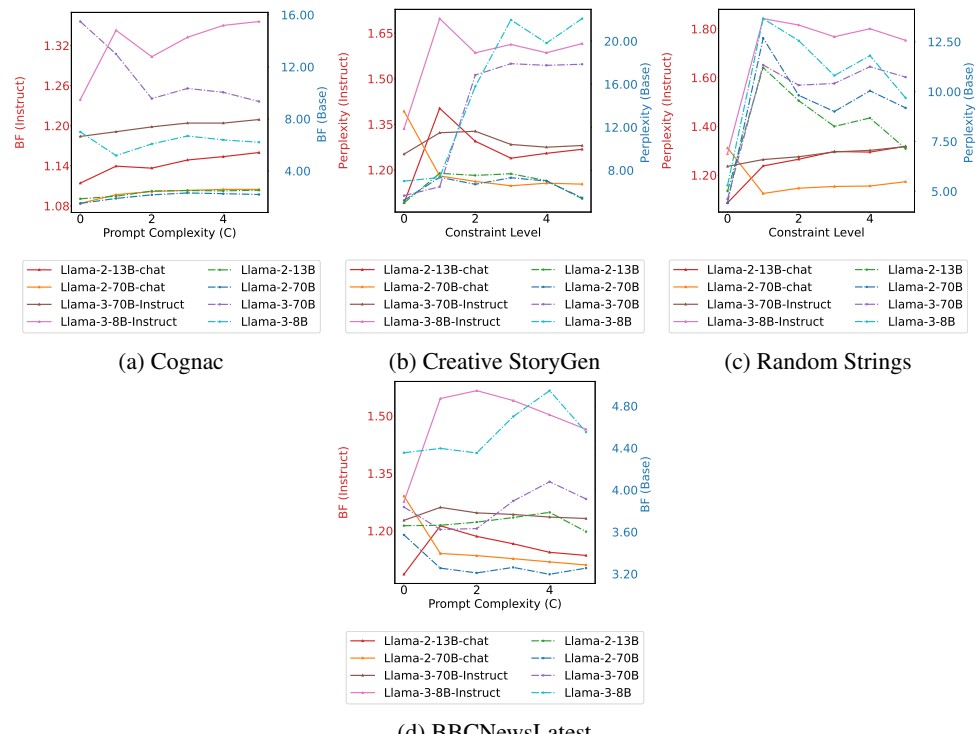

(a) Cognac     (b) Creative StoryGen     (c) Random Strings

(d) BBCNewsLatest

Figure 12: BF changes with prompt complexity ($C$) for Different Tasks. We can see prompt complexity affects BF in a task-varied way.

## G  GENERALIZATION TO ADDITIONAL TASKS AND MODELS

To confirm the generalizability of our findings (§ 5), we extend our experiments to new domains: summarization on XSUM (Narayan et al., 2018), multilingual tasks on AYA (Singh et al., 2024). We formulate prompt complexity $C$ as providing $C \times 25$ words in the prompt. We also verify our findings on a new model, Qwen3-4B (Team, 2025).[12] As presented in Figure 13, our core conclusions remain robust across these diverse conditions.

## H  PROOF OF LLM LOG-LIKELIHOOD CONVERGENCE

The following proof is a simplified version of the one in (Mudireddy et al., 2024), presented for completeness and to refine its original bounds. For the formal measure-theoretical treatment, we

---

[12]For the Qwen3 family, we use the Qwen3-4B-Base and Qwen3-4B-Instruct-2507 pair. Other aligned variants can be activated into a reasoning mode, exhibiting behavior distinct from the models in our main study, and were thus excluded for a fair comparison.

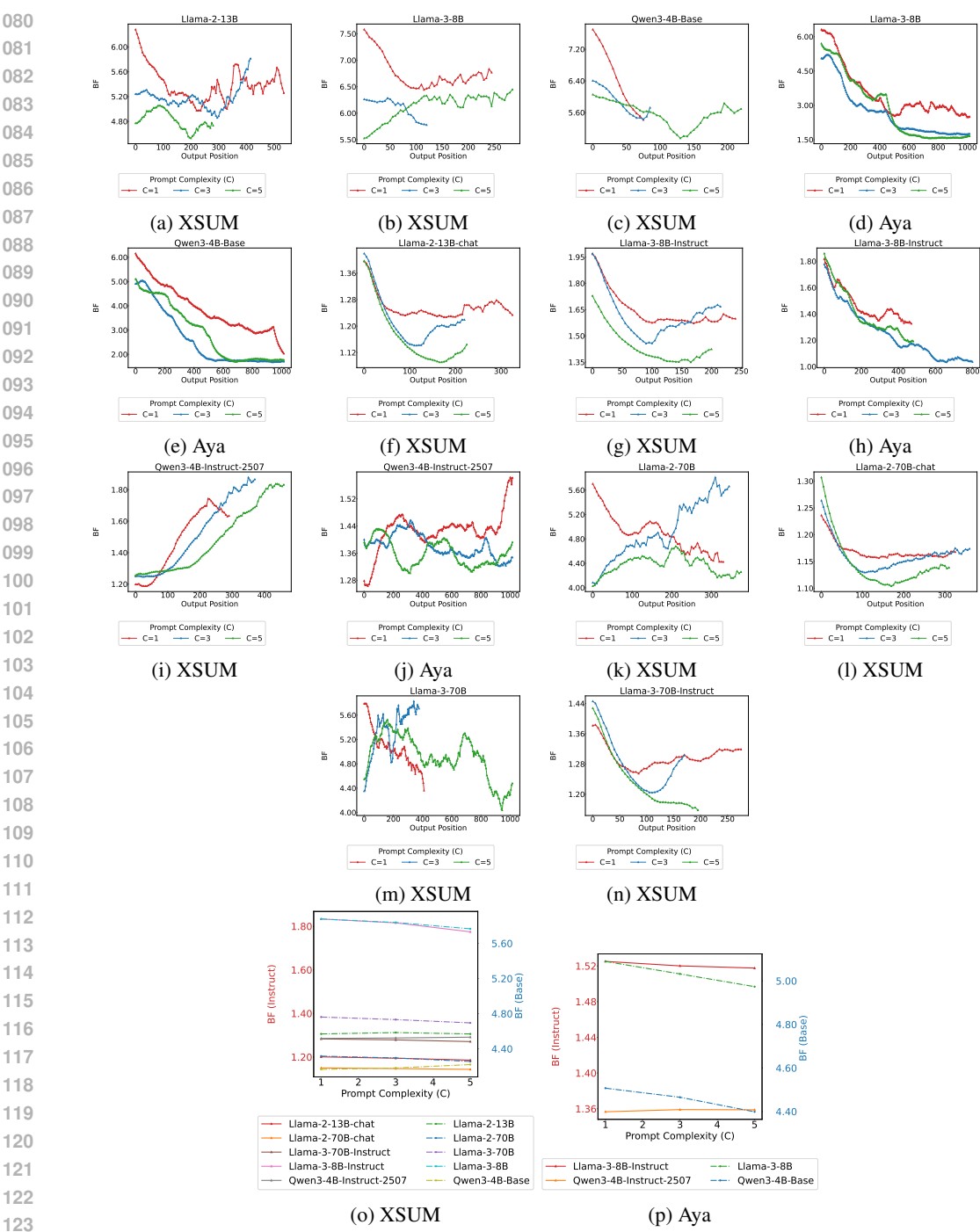

Figure 13: **Additional Verification on Summarization, Multilingual and Qwen Model family.** For better visualization, we compute the exponential moving averaged values of perplexity with the smoothing factor set as $0.1$.

refer readers to the original paper. While a more direct proof using the weak law of large numbers is possible, we use Chebyshev's inequality to provide a more self-contained and accessible argument.

The key observation here is that under current computation architecture, the probability implemented by transformers are log-precision (Merrill & Sabharwal, 2023), and thus $|\log P(y_{1:N}|x;\theta)|$ is bounded (e.g., $|\log P(y_{1:N}|x;\theta)| \leq M$). For the truncated probability $\tilde{P}(y_{1:N}|x;\theta)$, we can es-

sentially only consider the non-zero probability over the truncated vocabulary, and the same bound holds. Depending on the quantization scheme implemented, examples of $M$ include $32, 64$, etc.

We define the length-averaged *realized entropy* for a specific sequence $y_{1:N}$ as:

$$h_{\text{realized}}(y_{1:N}) \stackrel{\text{def}}{=} \frac{1}{N} \sum_{t=1}^{N} H(Y_t|[x, y_{<t}]; \theta) \tag{6}$$

where $H(Y_t|[x, y_{<t}]; \theta) = -\sum_{y \in V} P(y|[x, y_{<t}]; \theta) \log P(y|[x, y_{<t}]; \theta)$.

We aim to bound the probability that the NLL deviates from this realized entropy. Let $\Delta_N$ be the total difference between the log-probability and the realized entropy sum:

$$\Delta_N = (-\log P(y_{1:N}|x; \theta)) - \sum_{t=1}^{N} H(Y_t|[x, y_{<t}]; \theta) = \sum_{t=1}^{N} Z_t \tag{7}$$

where we define the single-step deviation variable $Z_t$ as:

$$Z_t \stackrel{\text{def}}{=} -\log P(y_t|[x, y_{<t}]; \theta) - H(Y_t|[x, y_{<t}]; \theta) \tag{8}$$

Note that $\Delta_N$ is a random variable formed by the sum of $Z_t$. To use Chebyshev's inequality, we need to calculate the variance of this sum:

$$\text{Var}(\Delta_N) = \text{Var}\left(\sum_{t=1}^{N} Z_t\right) = \sum_{t=1}^{N} \text{Var}(Z_t) + \sum_{i \neq j} \text{Cov}(Z_i, Z_j) \tag{9}$$

We now show that the covariance terms $\text{Cov}(Z_i, Z_j)$ are zero for all $i < j$. By definition, $\text{Cov}(Z_i, Z_j) = \mathbb{E}[Z_i Z_j] - \mathbb{E}[Z_i]\mathbb{E}[Z_j]$.

First, observe that the expected value of the deviation $Z_j$ at any step, conditioned on the prompt and generated history, is zero:

$$\mathbb{E}[Z_j|[x, y_{<j}]; \theta] = \mathbb{E}_{y_j}\left[-\log P(y_j|[x, y_{<j}]; \theta)\right] - H(Y_j|[x, y_{<j}]; \theta)$$
$$= H(Y_j|[x, y_{<j}]; \theta) - H(Y_j|[x, y_{<j}]; \theta) = 0 \tag{10}$$

This implies $\mathbb{E}[Z_j] = 0$ for all $j$. Thus, $\text{Cov}(Z_i, Z_j) = \mathbb{E}[Z_i Z_j]$.

For $i < j$, the value of $Z_i$ is fully determined by the history $[x, y_{<j}]$. We use the Law of Iterated Expectations, conditioning on the history up to step $j$:

$$\mathbb{E}[Z_i Z_j] = \mathbb{E}_{[x, y_{<j}]}\left[\mathbb{E}[Z_i Z_j|[x, y_{<j}]; \theta]\right] \tag{11}$$
$$= \mathbb{E}_{[x, y_{<j}]}\left[Z_i \cdot \mathbb{E}[Z_j|[x, y_{<j}]; \theta]\right] \quad \text{(since } Z_i \text{ is determined given } y_{<j}) \tag{12}$$
$$= \mathbb{E}_{[x, y_{<j}]}\left[Z_i \cdot 0\right] \quad \text{(by Eq. 10)} \tag{13}$$
$$= 0 \tag{14}$$

Since all cross-terms are zero, the variance of the sum is simply the sum of the variances:

$$\text{Var}(\Delta_N) = \sum_{t=1}^{N} \text{Var}(Z_t) \tag{15}$$

Given the observation that transformer probabilities are computed with bounded log-precision (Merrill & Sabharwal, 2023), we have $|\log P(y|[x, y_{<t}]; \theta)| \leq M$. Consequently, the random variable $Z_t$ is bounded, and its variance is bounded by a constant, denoted $C = (2M)^2$.

$$\text{Var}(\Delta_N) \leq N \cdot C \tag{16}$$

We can now apply Chebyshev's inequality to the length-averaged deviation:

$$P\left(\left|\frac{\Delta_N}{N}\right| \geq \epsilon\right) \leq \frac{\text{Var}(\Delta_N/N)}{\epsilon^2} = \frac{\frac{1}{N^2}\text{Var}(\Delta_N)}{\epsilon^2} \leq \frac{\frac{1}{N^2}(N \cdot C)}{\epsilon^2} = \frac{C}{N\epsilon^2} \tag{17}$$

Taking the limit as $N \to \infty$, the probability of deviation approaches 0. Thus, we have convergence in probability:

$$-\frac{1}{N} \log P(y_{1:N}|x; \theta) \xrightarrow{P} h_{\text{realized}}(y_{1:N}) \tag{18}$$

## I    FULL NUDGING EXPERIMENT RESULTS

Due to space limits, we put the nudging experiment results for MMLU here. Though on MMLU, nudging does not reduce BF that quickly as over Just-Eval-Instruct, it does bring down BF of base models significantly, which verifies our hypothesis in § 7.

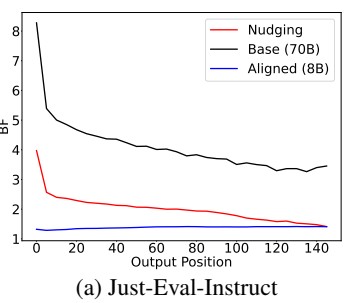
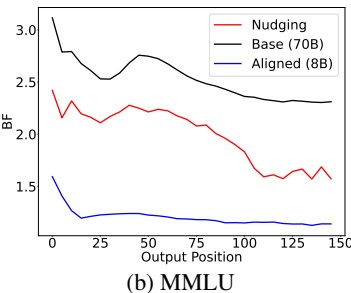

(a) Just-Eval-Instruct               (b) MMLU

Figure 14: Output Perplexity Dynamics in Nudging Experiments.

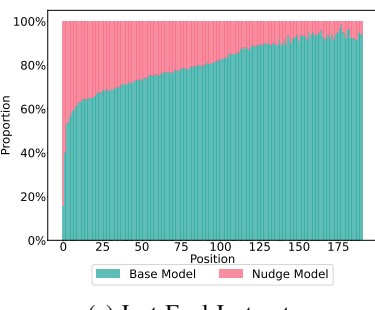
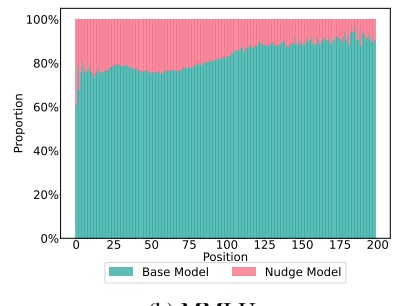

(a) Just-Eval-Instruct               (b) MMLU

Figure 15: Nudging Ratio Histogram.

## J    BF AND INFORMATION DENSITY

Our BF measure can also be interpreted as capturing the information density that LLMs target to facilitate efficient communication (Genzel & Charniak, 2002; Jaeger & Levy, 2006; Levy, 2008; Mahowald et al., 2013; Meister et al., 2021; Verma et al., 2023). Prior work has leveraged both token-level log-probabilities and entropy rates ($\bar{H}$) as proxies for information density in human and machine communication. In Theorem 4.1, we formalize the connection between these views, showing that BF–defined as the exponentiated entropy rate–aligns naturally with this theoretical framework. Unlike prior studies focused primarily on linguistic theory or cognitive science, our work operationalizes this principle at scale across modern LLMs, linking information density to alignment training, decoding dynamics, and output variability in a unified analysis.

## K    DISCUSSION: DIVERSITY AND BF CORRELATION

Following the branching factor (BF) analysis in § 2, a higher BF suggests greater lexical diversity in finite samples. To examine the relationship between BF and traditional diversity metrics, we compute Distinct-N (Li et al., 2016), incorporating necessary LLM-specific adaptations (Tevet & Berant, 2021; Guo et al., 2024; Kirk et al., 2024). We then conduct a correlation analysis between Distinct-N and BF.

Our results, presented in Figure 16, show **no consistent correlation** between BF and Distinct-N. Depending on the model and task, the relationship can be strongly positive, strongly negative, or entirely absent (e.g., Llama-3-70B-Instruct on Cognac at Figure 16b). This empirical inconsistency

highlights a fundamental conceptual point: *BF measures a property of the underlying probability distribution, whereas diversity metrics measure a surface property of finite samples.*

BF, as the exponentiated entropy, characterizes the "width" of the model's entire output distribution. In contrast, metrics like Distinct-N describe a small set of sampled outputs and are known to be unreliable proxies for distributional properties, being sensitive to confounding factors like generation length (Liu et al., 2022).[13] This distinction is critical, as two models can produce samples of similar diversity while having fundamentally different underlying distributions (e.g., with infinite KL-divergence), a nuance that BF captures but sample-based metrics miss. Therefore, our work focuses on probability concentration, measured by BF, as a more fundamental and insightful tool for understanding a model's generative process.

Viewing alignment through the lens of BF reduction provides a unified framework that explains several disparate observations: it clarifies how alignment shrinks the generative horizon, why aligned models are less sensitive to decoding methods, and how techniques like Chain-of-Thought stabilize generation by shifting information to low-BF regions. This focus on distributional properties aligns with emerging research highlighting the importance of a model's entropy in understanding and improving advanced reasoning capabilities Cui et al. (2025); Wu et al. (2025b).

## L  CONFOUNDER INVESTIGATION: DATA CONTAMINATION

A potential confounder in our analysis is the influence of data contamination. If prompts closely resemble the training data (including pretraining and alignment tuning, i.e., "data contamination"), smaller BF values would be expected, and vice versa. To evaluate this, we use the Min-K% metric (Shi et al., 2024b), which quantifies the overlap between experimental prompts and training data. Following Shi et al. (2024b), we set $K = 20$ and compute the average log-likelihood for the minimum $K\%$ of tokens. Using these Min-K% values, we perform a linear regression with BF to assess their correlation. For each task-model pair, Signed $R^2$ values are reported to indicate the strength and sign (positive or negative) of the correlation.

The results of the Min-K% analysis are presented in Figure 17. Significant negative correlations between BF and Min-K% are observed for models such as Llama-3-8B-Instruct, Llama-3-70B-Instruct, and Llama-2-70B-Chat across several tasks. Conversely, Llama-3-8B and Llama-2-13B-Chat models exhibit positive correlations. For other models, correlations are notably weaker. Overall, there is no consistent correlation pattern between BF and Min-K% across datasets and models, suggesting that data contamination cannot fully explain our findings.

---

[13]While the EAD metric (Liu et al., 2022) mitigates this issue, it remains influenced by vocabulary size and is not model-agnostic.

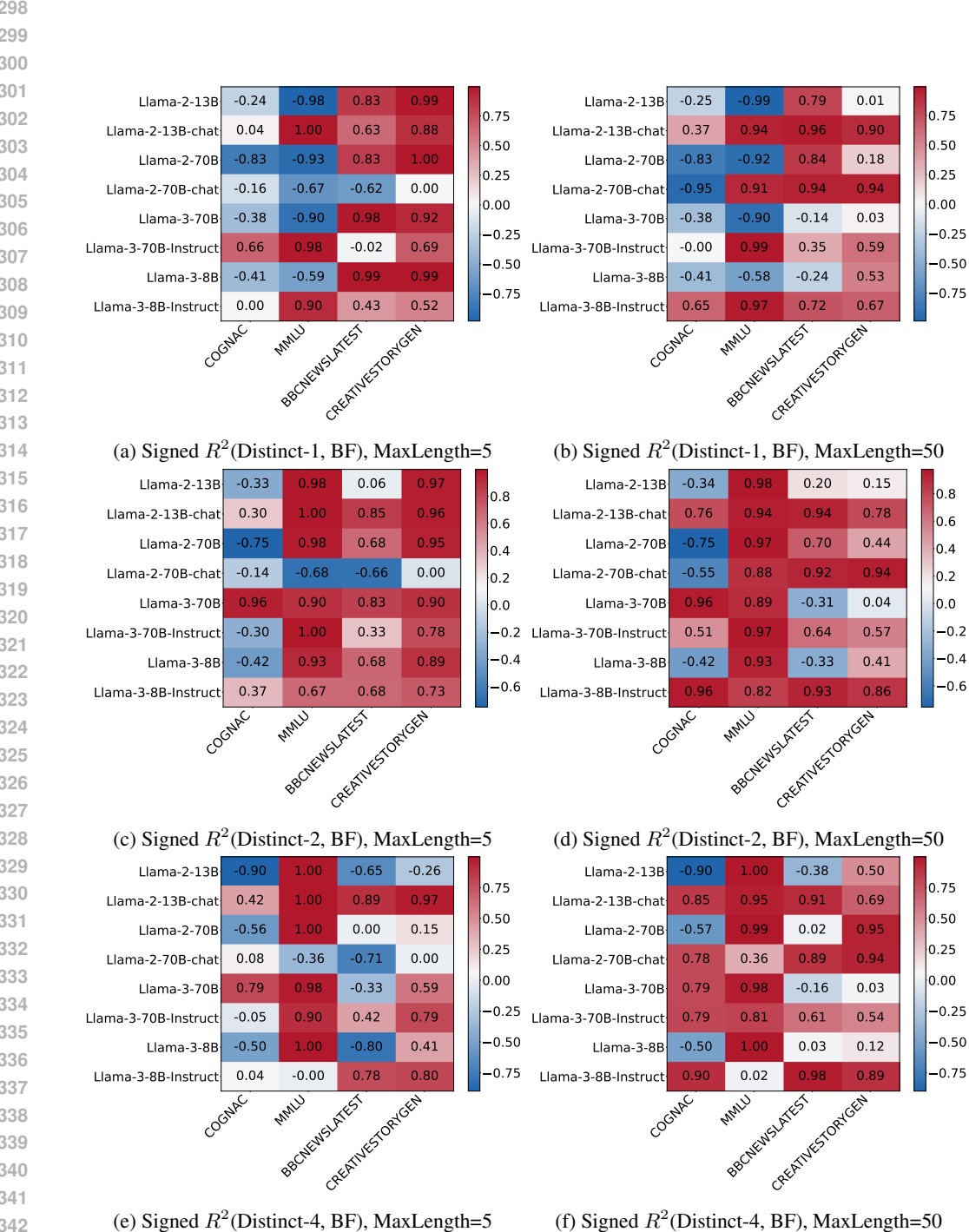

(a) Signed $R^2$(Distinct-1, BF), MaxLength=5

(b) Signed $R^2$(Distinct-1, BF), MaxLength=50

(c) Signed $R^2$(Distinct-2, BF), MaxLength=5

(d) Signed $R^2$(Distinct-2, BF), MaxLength=50

(e) Signed $R^2$(Distinct-4, BF), MaxLength=5

(f) Signed $R^2$(Distinct-4, BF), MaxLength=50

Figure 16: Correlational Analysis of BF and Distinct-N. We can find there is no consistent correlation between Distinct-N and BF.

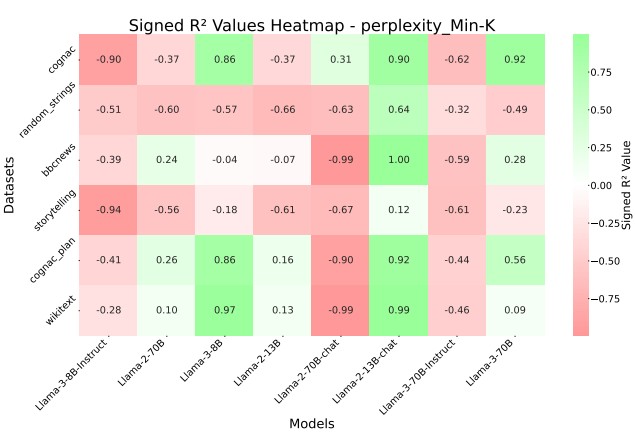

Figure 17: Signed $R^2$ values heatmap investigating correlation between EBF and Min-K %.

