# OpenReview forum: "LLM Probability Concentration: How Alignment Shrinks the Generative Horizon"
_ICLR.cc/2026/Conference — Submitted to ICLR 2026_

### Official Review · Reviewer_fwb1 · 2025-10-30

**Soundness:** 1
**Presentation:** 2
**Contribution:** 3
**Rating:** 2
**Confidence:** 3

**Summary:**

This work explores the behavior of LLMs during generation, focusing on the contrast between base models and aligned models that have undergone post-training. The authors propose a metric named "branching factor" (BF) to quantify the model probability distribution during generation, where BF represents how wide or narrow is the the space of plausible paths to continue the generation, and is closely related to perplexity measures. In a set of experiments over several models and datasets, they show that aligned models have a much lower BF, reflecting a narrower space of high-probability outputs, and connect this to the generation behavior where aligned LLMs are less sensitive to different decoding methods. They also present some analyses on the effect of intervening in the generation process at different stages.

**Strengths:**

1. A lot of the experimental results here are not trivial, and help shed some light on what the behavior of aligned vs. non-aligned models looks like.
2. Extensive experimental results in the paper including detailed appendices.
3. The subject matter would likely appeal to a broad community.

**Weaknesses:**

1. In a few places I felt there is a bit of conflation between hypotheses/conjecture and things that are actually demonstrated empirically in the experiments. Specifically:
- In Table 2, the results show that the aligned models have lower standard deviation in task accuracy. This result does not in itself "confirm that BF is a reliable predictor of sampling consistency" (l. 377). What we see is a very anecdotal pattern of correlation between BF and accuracy STD, we do not know that this pattern is reliable and consistent (especially given the very small absolute differences in BF), and we certainly don't know that there is a causal connection between these factors (as opposed to, say, a general connection between higher task performance and lower STD).
- In Fig. 5, we see the effect of the resampling intervention. This is interesting in itself, but again I don't see how it can be used to directly infer that the BF metric "reflects a deeper commitment to specific generative paths", just because BF and the output token index are correlated.
- In §7 and also in the intro and abstract there is talk about the effect of stylistic tokens, but I did not see that this is demonstrated anywhere in the paper.
2. The main focus of the paper is on the BF metric, and in their motivation the authors state that "token-level metrics such as entropy or log-likelihood… offer only a narrow lens on model behavior: they capture local properties but miss the global structure of the output space… this motivates our proposal of the BF". As the BF is a major contribution I would have expected the paper to demonstrate this statement in some way - directly comparing how the conclusions on model behavior they draw from BF in this work differ (if at all) from the conclusions reached in prior works that supposedly focused on more "local" properties. Moreover, in practice the calculation is very similar to mean token entropy and it wasn't entirely clear to me that BF really measures something different.
3. I found some later parts of the paper harder to follow and less self-contained - I was missing a more technical description of how exactly the resampling in §6 was performed, and I felt the concept of "nudging" (§7) and the motivation behind it was not sufficiently clear from the text.
4. It is worth mentioning that the conclusions in §3 have largely been shown by prior work, most explicitly in Shi et al. 2024 "A Thorough Examination of Decoding Methods in the Era of LLMs" (https://aclanthology.org/2024.emnlp-main.489/).

**Questions:**

1. Is there a reason for the mix between 70B and 8B models in §7? This combines the factor of large/small and aligned/not-aligned which may make the results more difficult to interpret.
2. Why does the text reference an average of "ten times higher BF" (line 335)? As far as I can see in most of the panels in Fig. 3 / Fig. 8 the ratio is much lower.


Minor suggestions:
* I did not feel the extra y-axis (cumulative impact) in Fig. 4 adds information to the reader, same for the 80% threshold (and it is unclear from the text why this threshold is important)
* The related work section focuses a lot on semantic entropy which IMO is a bit further removed from this work; in contrast, I think elaborating more on some works mentioned in the intro could provide more context to this work - specifically the various works mentioned under Hypotheses 1-3 (l. 183-189).

Typos:

l. 360 model size M -> model size S

l. 362 gains the use -> gains from the use

l. 450 - which applies

---

> ### Author Response · Authors · 2025-11-21
> **Responses to Reviewer fwb1 [1/3]**
>
> We thank Reviewer fwb1 for the detailed and careful review. We appreciate the opportunity to clarify the interpretation of our experiments and the positioning of our work.
>
> > **W1: In a few places I felt there is a bit of conflation between hypotheses/conjecture and things that are actually demonstrated empirically in the experiments. Specifically:**
> > **W1a:** In Table 2, the results show that the aligned models have lower standard deviation in task accuracy. This result does not in itself "confirm that BF is a reliable predictor of sampling consistency" (l. 377). What we see is a very anecdotal pattern of correlation between BF and accuracy Std, we do not know that this pattern is reliable and consistent (especially given the very small absolute differences in BF), and we certainly don't know that there is a causal connection between these factors (as opposed to, say, a general connection between higher task performance and lower Std).
>
> **R1a:** We appreciate this distinction. We agree that our language should be precise: our use of "predictor" was intended to denote a strong statistical correlation rather than a causal mechanism. However, we respectfully disagree that the observed pattern is "anecdotal." First, we have verified aligned models exhibits far more smaller BF than their base counterparts consistently across multiple model families (Llama-2, Llama-3 and Qwen), model sizes (8B, 70B), and diverse tasks. Second, prior extensive studies on decoding dynamics [1, 2, 3] have independently established that aligned models exhibit lower output variance than base counterparts. When viewed in the context of this established literature, the link between BF and consistency is well-supported. Third, regarding the "small absolute differences": We utilized near-greedy sampling ($T=0.6, p=0.9$) as a stress test. This naturally concentrates probability and compresses BF values; the fact that we still observe distinct, consistent differences between base and aligned models under these conditions reinforces the sensitivity of the metric. **To explicitly address the concern regarding reliability given these small values, we have since conducted a bootstrapping analysis (100 runs) on the results in Table 2. This analysis reveals that the fluctuation in BF values is negligible ($\approx 0.01$), confirming that the reported differences between base and aligned models are robust and statistically distinct, despite the compressed scale.**
>
> While this does not prove causation, it provides strong evidence that BF captures a fundamental aspect of the model's distributional determinism that is directly relevant to output consistency. To address your valid concern and strengthen the claim, we will revise the text to explicitly frame the relationship as a strong correlation, not a causal link.
>
> > **W1b:** In Fig. 5, we see the effect of the resampling intervention. This is interesting in itself, but again I don't see how it can be used to directly infer that the BF metric "reflects a deeper commitment to specific generative paths", just because BF and the output token index are correlated.
>
> **R1b:** We thank the reviewer for the question, which gives us an opportunity to clarify our reasoning.  Our reasoning is as follows:
> 1.  Our prior results establish that BF consistently decreases as generation proceeds (the model becomes more certain).
> 2.  The resampling experiment shows that intervening at later, lower-BF positions causes a much sharper drop in performance than intervening at earlier, higher-BF positions.
>
> Our results show that the outcome of this intervention (the drop in performance) is strongly predicted by the BF at the point of intervention. When the model is in a low-BF state (which tends to occur later in generation), the alternative paths are not just less probable, but of lower quality. This provides direct, functional evidence that the probability concentration measured by BF corresponds to a "deeper commitment to specific generative paths," where deviation is more costly. We will revise the paper to state this argument more clearly. To ensure a fair comparison across different stages of generation, our analysis is carefully conditioned on the token index, which avoids confounding generation progress with total output length.

---

> ### Author Response · Authors · 2025-11-21
> **Responses to Reviewer fwb1 [2/3]**
>
> > **W1c:** In §7 and also in the intro and abstract there is talk about the effect of stylistic tokens, but I did not see that this is demonstrated anywhere in the paper.
>
> **R1c:** Thank you for asking for clarification. This is demonstrated in the **nudging experiments** in Section 7. In these experiments, we take a base model and prefix its generation with a short, stylistic phrase generated by an *aligned* model (e.g., "Sure, here is..."). We show that this "nudge" is sufficient to shift the base model into a lower-BF generation regime, similar to that of a fully aligned model. This provides direct evidence for our hypothesis that alignment works in part by steering the model towards these specific, low-entropy conversational patterns. We have revised Section 7 to make the role of these "stylistic tokens" in the experimental design more explicit.
>
> > **W2:** The main focus of the paper is on the BF metric, and in their motivation the authors state that "token-level metrics such as entropy or log-likelihood… offer only a narrow lens on model behavior: they capture local properties but miss the global structure of the output space… this motivates our proposal of the BF". As the BF is a major contribution I would have expected the paper to demonstrate this statement in some way - directly comparing how the conclusions on model behavior they draw from BF in this work differ (if at all) from the conclusions reached in prior works that supposedly focused on more "local" properties. Moreover, in practice the calculation is very similar to mean token entropy and it wasn't entirely clear to me that BF really measures something different.
>
> **R2:** You are correct that BF is formally defined as the exponentiated length-averaged entropy. The core contribution of our work is not to propose a novel mathematical formula, but to introduce a **unifying conceptual framework**.
>
> Our central argument is that the question, "What is the effective branching factor of an LLM?", provides a powerful lens for understanding a range of behaviors. The value of this approach lies in its ability to connect disparate phenomena under a single, coherent explanation. Specifically, the BF framework allows us to quantitatively link:
> 1. Alignment tuning reduces BF → shrinks the generative horizon
> 2. Low BF explains decoding sensitivity → fewer viable options to prune
> 3. CoT stabilizes generation → shifts key information to late, low-BF regions
> 4. Avoid late branching → too many low-probability, low-quality continuations
> 5. Alignment surfaces low-entropy paths already latent in base models
>
> While other metrics can measure aspects of diversity, they are often sensitive to surface-level factors (like vocabulary size) and do not offer the same consistent, distribution-level explanation across different models and tasks. Our focus is on diagnosing the fundamental properties of the model's output distribution, and the BF framework provides the right tool for this question. We will clarify this positioning in the introduction.
>
> > **W3:** I found some later parts of the paper harder to follow and less self-contained - I was missing a more technical description of how exactly the resampling in §6 was performed, and I felt the concept of "nudging" (§7) and the motivation behind it was not sufficiently clear from the text.
>
> **R3:** Thank you for this feedback on clarity. We clarify technical details for both experiments here:
>
> - **Resampling (Section 6):** We have added a detailed algorithmic description in the revision: at position $t$, we take the prefix $y_{1:t-1}$ generated so far, then sample a new continuation $y'\_{t:N}$ from the model's distribution $P(Y_{t:N} | x, y_{1:t-1})$. We then evaluate the full sequence $[y_{1:t-1}, y'_{t:N}]$ on the task.
>
> - **Nudging (Section 7):** In the revision, we have clarified: (1) the motivation (testing whether alignment surfaces latent low-entropy paths); (2) the procedure (using an aligned model to generate prefixes (as in [1]), then continuing with a base model); (3) what we mean by "nudging" (providing stylistic tokens that steer generation); and (4) how this relates to our hypothesis about alignment (how does alignment tuning impact BF? ).

---

> ### Author Response · Authors · 2025-11-21
> **Responses to Reviewer fwb1 [3/3]**
>
> > **W4:** It is worth mentioning that the conclusions in §3 have largely been shown by prior work, most explicitly in Shi et al. 2024 "A Thorough Examination of Decoding Methods in the Era of LLMs" (https://aclanthology.org/2024.emnlp-main.489/).
>
> **R4:** Thank you for the reference. We agree entirely. As indicated by the section title, "Case Study," Section 3 is intended to motivate our investigation by presenting a well-known phenomenon: the reduced sensitivity of aligned models to decoding methods. Our contribution is not to report this observation as novel, but to provide a deeper, quantitative explanation for *why* it occurs. The BF framework shows that this insensitivity is a direct consequence of the severe probability concentration induced by alignment. We add a citation to Shi et al. (2024) in Section 3 and clarify the role of this section as a motivating example for the deeper analysis that follows.
>
> > **Q1:** Is there a reason for the mix between 70B and 8B models in §7? This combines the factor of large/small and aligned/not-aligned which may make the results more difficult to interpret.
>
> **A1:** This was a deliberate choice to isolate the effect of the "nudge." We use a small aligned model (8B-Instruct) to generate the initial stylistic prefix and a large base model (70B) for the main generation. The goal is to show that even a very small, aligned-style intervention can steer a much larger base model into a low-BF trajectory. This demonstrates that the effect is not about the raw capabilities of the aligned model, but about the powerful steering effect of the stylistic patterns it learns. We clarify the motivation for this experimental design in Section 7.
>
> > **Q2:** Why does the text reference an average of "ten times higher BF"?
>
> **A2:** This figure comes from comparing the typical BF values we observe for base models (often around 12) with those for aligned models (around 1.2). While the exact ratio varies by task and model, this "order of magnitude" difference is a frequent pattern in our results. We clarify that this is a representative average to avoid confusion in Section 5.1.
>
> ### Minor Suggestions
>
> Thank you for catching these typos and suggestions. We will:
> - Fix the typo on l. 360: "model size M" → "model size S"
> - Fix the typo on l. 362: "gains the use" → "gains from the use"
> - Fix the typo on l. 450: add "which applies"
> - Clarify the extra y-axis in Fig. 4
> - Clarify why the 80% threshold is useful
>
> ### References
>
> [1] Song, Yifan, et al. "The good, the bad, and the greedy: Evaluation of llms should not ignore non-determinism." Proceedings of the 2025 Conference of the Nations of the Americas Chapter of the Association for Computational Linguistics: Human Language Technologies (Volume 1: Long Papers). 2025.
>
> [2] Renze, Matthew. "The effect of sampling temperature on problem solving in large language models." Findings of the association for computational linguistics: EMNLP 2024. 2024.
>
> [3] Shi, Chufan, et al. "A Thorough Examination of Decoding Methods in the Era of LLMs." Proceedings of the 2024 Conference on Empirical Methods in Natural Language Processing. 2024.
>
> [4] Fei, Y., Razeghi, Y., & Singh, S. (2024). Nudging: Inference-time Alignment of LLMs via Guided Decoding. arXiv preprint arXiv:2410.09300.
>
> [5] Shi, W., et al. (2024). A Thorough Examination of Decoding Methods in the Era of LLMs. In Proceedings of EMNLP 2024.

---

### Official Review · Reviewer_nL97 · 2025-10-31

**Soundness:** 4
**Presentation:** 4
**Contribution:** 3
**Rating:** 6
**Confidence:** 3

**Summary:**

This paper tackles the phenomenon that aligned LLMs produce outputs that are significantly less diverse than their base model counterparts. The authors introduce a metric they term the Branching Factor (BF) to formalize and measure this "probability concentration".

The central thesis is that alignment tuning acts as a "shrinking" mechanism on the model's generative horizon, and the paper provides a strong, unified framework that connects this concentration to several other observed behaviors, such as decoding insensitivity and the stability of CoT reasoning.

The paper's core contribution is the formalization and application of the Branching Factor. Unlike perplexity, BF is measuring the perplexity of the space the model chooses to explore on its own. With this, the authors find that alignement is the dominant factor reducing from ~12 to ~1.2 going from base model to instruction tuned models. Also, the generation locks into a specific topic or reasoning and becomes more predictable as it generates more tokens, which is verified with the resampling experiment and the nudging experiment.

**Strengths:**

1. **Unifying Metric**: The paper's primary strength is providing a single, intuitive metric, BF, that explains and unifies several disparate, known phenomena: low diversity, insensitivity to decoding parameters, and the stability of CoT reasoning.
2. **Strong Experimental Design:** The resampling and nudging experiments are clever and highly effective ways to demonstrate the consequence of BF reduction, showing that models become "brittle" and locked into their chosen path.
3. **Robustness and Clarity:** The findings are shown to be consistent across multiple model families and various tasks. The Pareto analysis is particularly effective at isolating alignment as the key variable.

**Weaknesses:**

1. **Metric Novelty:** The paper proposes "Branching Factor," but it is almost a specific measurement of perplexity.
2. **Complexity in alignment:** The authors mentioned that they did not disentangle which stage of
alignment contributes most to BF reduction, and ideally some experiment on this should be in the paper.

**Questions:**

1. How do you consider beam search as part of the picture, i.e. do you think that since "off-path" trajectories in aligned models are not just low-probability but low-quality, are alternative beams are likely to be "garbage" paths?

---

> ### Author Response · Authors · 2025-11-21
> **Responses to Reviewer nL97**
>
> We thank Reviewer nL97 for the positive feedback and for recognizing the value of our unified framework and strong experimental design. We appreciate the opportunity to clarify our contributions and the scope of our work.
>
> > **W1: Metric Novelty:** The paper proposes "Branching Factor," but it is almost a specific measurement of perplexity.
>
> **R1:** We agree that BF is formally equivalent to the information-theoretic definition of perplexity (i.e., exponentiated entropy). Our contribution is not the invention of a new metric, but rather the development of a **unifying conceptual framework** built upon it.
>
> As you kindly noted in your review, the paper's primary strength is in providing a single, intuitive metric that "explains and unifies several disparate, known phenomena." The novelty and value of our work lie in demonstrating that the lens of probability concentration, as quantified by BF, can connect:
> 1. Alignment tuning reduces BF → shrinks the generative horizon
> 2. Low BF explains decoding sensitivity → fewer viable options to prune
> 3. CoT stabilizes generation → shifts key information to late, low-BF regions
> 4. Avoid late branching → too many low-probability, low-quality continuations
> 5. Alignment surfaces low-entropy paths already latent in base models
>
>
> By framing the analysis around the question, "What is the effective branching factor?", we provide an intuitive and powerful tool for understanding *how* alignment reshapes the model's output distribution. We will refine our introduction in revision to make it clearer that our contribution is this unifying framework, not the metric itself.
>
> > **W2: Complexity in alignment:** The authors mentioned that they did not disentangle which stage of alignment contributes most to BF reduction, and ideally some experiment on this should be in the paper.
>
> **R2:** This is an excellent point and a key direction for future research. We acknowledge this as a limitation in our Discussion section. Disentangling the precise contribution of each alignment stage is practically challenging due to two main factors:
> 1.  **Limited Checkpoint Availability:** Most open-source model releases only provide the final base and aligned versions, without the intermediate checkpoints from SFT, reward modeling, and RL stages.
> 2.  **Iterative Alignment Processes:** As detailed in technical reports for models like Llama 3 [1], modern alignment is not a simple linear process but often involves multiple, iterative cycles of training and data curation, blurring the lines between discrete "stages."
>
> While a full-scale ablation is beyond the scope of this work due to these constraints, we do cite concurrent work (e.g., [2, 3]) that investigates entropy dynamics during RL, suggesting that the reinforcement learning phase is a primary driver of this concentration. We are actively exploring this with models where more checkpoints are available. We include preliminary results from this analysis on OLMo2 series in the revision (Section 7) to provide a more detailed dissection.
>
> > **Q1:** How do you consider beam search as part of the picture, i.e. do you think that since "off-path" trajectories in aligned models are not just low-probability but low-quality, are alternative beams are likely to be "garbage" paths?
>
> **A1:** This is a great question. Our findings strongly suggest that for aligned models, this is often the case. Our resampling experiment (Section 6) provides direct evidence: when we force the model off its highest-probability path, especially in later, low-BF stages of generation, task performance drops sharply. This implies that it's easier to obtain lower-quality samples when probability is already highly concentrated.
>
> For a highly aligned model with a BF close to 1.2, there is very little probability mass distributed among alternative paths. Consequently, beam search has little to work with; the alternative beams are likely exploring parts of the distribution that the alignment process has effectively pruned for being undesirable. In contrast, for a base model with a much higher BF, beam search has a richer set of plausible alternatives to explore.
>
> This explains why decoding methods often have a limited impact on aligned models—the die is already cast by the model's highly concentrated distribution. We will incorporate this point into our discussion to make the practical implications for decoding strategies like beam search clearer.
>
> ### References
>
>
> [1] Dubey, A., Jauhri, A., Pandey, A., Kadian, A., Al-Dahle, A., Letman, A., ... & Ganapathy, R. (2024). The llama 3 herd of models. arXiv e-prints, arXiv-2407.
>
> [2] Cui, G., Zhang, Y., Chen, J., Yuan, L., Wang, Z., Zuo, Y., ... & Ding, N. (2025). The entropy mechanism of reinforcement learning for reasoning language models. arXiv preprint arXiv:2505.22617.
>
> [3] Wu, F., Xuan, W., Lu, X., Harchaoui, Z., & Choi, Y. (2025). The Invisible Leash: Why RLVR May Not Escape Its Origin. arXiv preprint arXiv:2507.14843.

---

> > ### Comment · Reviewer_nL97 · 2025-11-26
> >
> > Thank you for the detailed reply. I will keep my score.

---

### Official Review · Reviewer_fLap · 2025-11-03

**Soundness:** 3
**Presentation:** 2
**Contribution:** 3
**Rating:** 4
**Confidence:** 4

**Summary:**

The paper proposes the Branching Factor (BF), a scalar metric grounded in information theory that measures how many plausible next tokens a language model entertains at each generation step. By expressing BF in terms of per‐token entropy and invoking the Asymptotic Equipartition Property, the authors derive an estimator that scales to realistic sequence lengths without exhaustive enumeration. They evaluate BF across base, instruction‐tuned, and Chain‐of‐Thought variants of contemporary LLMs on tasks ranging from summarization to multi‐step reasoning, observing that BF declines steadily as generation proceeds and that alignment tuning collapses BF by almost an order of magnitude from the very first token. Nudging experiments indicate that alignment does not fundamentally rewrite model parameters but rather surfaces latent low‐entropy trajectories present in the unaligned model. Overall, the work offers both theoretical insight and practical diagnostics for understanding the reduction in output diversity that often accompanies model alignment.

**Strengths:**

The paper introduces a clear, distribution-level lens on why aligned LLMs tend to be more deterministic, formalizing “probability concentration” via a task-agnostic Branching Factor (BF) instead of surface diversity metrics. The BF is grounded in information theory—defined as the exponentiated entropy rate over continuations—and connected to a balanced-tree abstraction of the effective output space, which makes the idea intuitive and comparable across settings. The authors also provide two practical estimators: a token-entropy aggregation for short outputs and an AEP-based estimator for long outputs that leverages length-averaged NLL, with empirical plots showing NLL closely tracks entropy and stabilizes with length. This framework unlocks several cohesive empirical findings: BF typically shrinks as generation proceeds; aligned models sit near BF≈1.2 (roughly an order of magnitude lower than base models), helping explain reduced decoding sensitivity; and a Pareto-style analysis highlights alignment as the dominant driver relative to model size, generation, and prompt complexity. Beyond diagnostics, the work ties BF to behavior: majority-vote variance drops with lower BF, resampling late in a sequence degrades performance more than early resampling, and CoT’s longer chains naturally push inference into low-BF regions that stabilize answers. Together, the formalization, efficient estimation, and multi-angle evidence (decoding study, variance analyses, resampling, and nudging) create a persuasive, unified account of alignment’s impact on LLM outputs.

**Weaknesses:**

Some claims hinge on estimator assumptions and experimental choices that invite further stress-testing. The AEP-based estimator inherits conditions (e.g., long sequences, autoregressive generation, finite precision) and approximations; while these are argued to be mild and empirically supported, deviations (short outputs, atypical decoding, domain shift) could bias BF estimates, and Monte Carlo underestimation issues remain salient for short generations. Causal attributions around alignment are suggestive rather than surgical: alignment dominates in the Pareto analysis, but the study does not disentangle which alignment stage (SFT vs. reward modeling vs. RL) drives BF reductions; the authors themselves flag this and hypothesize RL as the main culprit, leaving an important gap for checkpoint-level ablations. Some demonstrations (e.g., nudging with a different “instruct” model for prefixes) risk confounds from model mixing; similarly, results are concentrated on specific open-weight families and tasks, so generalization to other architectures, languages, and safety/creative settings deserves replication. Finally, while BF is positioned as deeper than sample-level diversity, its practical relationship to user-perceived variety is complex and sometimes uncorrelated with Distinct-N; practitioners may still need guidance on how BF should inform decoding or training interventions without sacrificing quality. These caveats don’t undercut the paper’s core insight, but they do mark clear avenues for more rigorous causal analyses, broader model/task coverage, and tooling that translates BF diagnostics into actionable training or deployment knobs.

**Questions:**

The manuscript would be strengthened by deeper engagement with related and contrasting work. Just to name a few, the recent paper by Rodemann et al. https://arxiv.org/abs/2502.14581 exploring implicit statistical biases via alignment deserve explicit discussion. Contra‐alignment perspectives such as the Overton Pluralism framework proposed by Lake et al. (https://arxiv.org/abs/2406.17692) are mentioned, but could be discussed in more detail. By omitting these debates, the paper risks overstating novelty and understating the broader scholarly context. On the methodological side, the assumptions of ergodicity and stationarity required for AEP may be violated in highly variable prompts - open‐domain dialogue or code synthesis, for example - and quantitative error bounds under such conditions would bolster the argumetn. While I cannot vote for acceptance of the paper in its current form, I am confident that a thorough revision of the currently very brief related works (sec 8) section and properly addressing my concerns wrt ergodictiy and stationarity can substantially improve the paper and lift it over the bar.

---

> ### Author Response · Authors · 2025-11-21
> **Responses to Reviewer fLap [1/2]**
>
> We thank Reviewer fLap for the detailed and constructive feedback. We are glad you found that our work offers a clear, distribution-level lens on probability concentration, supported by a cohesive set of empirical findings. We appreciate the opportunity to address your concerns regarding the scope of our work and its relation to the broader literature.
>
> > **W1:** Some claims hinge on estimator assumptions and experimental choices that invite further stress-testing. The AEP-based estimator inherits conditions (e.g., long sequences, autoregressive generation, finite precision) and approximations; while these are argued to be mild and empirically supported, deviations (short outputs, atypical decoding, domain shift) could bias BF estimates, and Monte Carlo underestimation issues remain salient for short generations.
>
> **R1:** We thank the reviewer for the detailed questions. We address them point-by-point:
> 1.  **Short vs. Long Sequences:** You are correct that the best estimator depends on sequence length. As we state in the paper (Sec 4.1, Footnote 6 in previous version, L248 for Section 4 in the revision), for **short sequences** where AEP guarantees are weaker, we use a direct Monte Carlo (MC) estimation of the per-token entropy. We show empirically (Fig 2 and Appendix C) that this provides stable estimates with a reasonable number of samples. For **long sequences**, where direct MC systematically underestimates entropy, the AEP provides a much more reliable estimator via the length-averaged log-likelihood. Our methodology explicitly handles both regimes appropriately.
> 2.  **Atypical Decoding:** Our theoretical assumptions are robust to different decoding strategies. The AEP for LLMs only requires that the model be **autoregressive**—that is, the sequence probability factorizes into a product of next-token probabilities. This property holds for standard LLMs regardless of whether decoding is greedy, top-p, or uses other common samplers. Decoding strategy affects *which path* is taken through the distribution, but not the validity of the underlying probabilistic framework.
> 3.  **Domain Shift:** We agree that BF values are domain-dependent, as the model's output distribution is conditioned on the prompt. This is an expected property, not a flaw in the estimator. We intentionally tested our framework on a **highly diverse set of tasks** to demonstrate its general applicability precisely because of this. Our experiments span creative open-ended generation (Storytelling), controlled QA (Cognac), reasoning (MMLU), summarization (XSUM, Appendix G), multilingual generation (Aya, Appendix G), and even an extreme out-of-domain case (Random Strings). The consistent patterns we observe across these varied domains strengthen our conclusions about the fundamental effects of alignment.
>
> > **W2:** Causal attributions around alignment are suggestive rather than surgical: alignment dominates in the Pareto analysis, but the study does not disentangle which alignment stage (SFT vs. reward modeling vs. RL) drives BF reductions; the authors themselves flag this and hypothesize RL as the main culprit, leaving an important gap for checkpoint-level ablations.
>
> **R2:** We agree that our analysis identifies alignment tuning as the dominant *factor* among those we compared, but it does not surgically disentangle the precise contribution of each sub-stage. We state this clearly as a limitation in the submission version. Disentangling these stages is a significant research challenge in itself, primarily due to the lack of publicly available intermediate checkpoints for most models and the complex, iterative nature of modern alignment pipelines (as described in the Llama 3 technical report [1]). However, as we note for other reviewers, we are actively exploring this with the OLMo model series, which offers more granular checkpoints, and we include the initial findings in the revision (Section 7, "Which Training Stage Reduces BF Most?"). Our paper provides the foundational diagnostic framework that makes such deeper investigations possible.

---

> ### Author Response · Authors · 2025-11-21
> **Responses to Reviewer fLap [2/2]**
>
> > **W3:** Some demonstrations (e.g., nudging with a different “instruct” model for prefixes) risk confounds from model mixing; similarly, results are concentrated on specific open-weight families and tasks, so generalization to other architectures, languages, and safety/creative settings deserves replication.
>
> **R3:** We address these two points separately:
> *   **Nudging Experiment:** The use of a smaller, different model for the "nudge" is a deliberate methodological choice, following the original Nudging paper [2]. The goal is to test the hypothesis that a small number of "stylistic tokens" are sufficient to steer a much more powerful base model onto a low-entropy path. Using a small model for the prefix demonstrates that this effect is not about the raw capability of the prefix-generating model, but about the powerful steering function of the tokens themselves. We have made this explicit in L448, Section 7 in the revision.
> *   **Generality:** We respectfully suggest our findings are more general than portrayed. On the model side, we confirm our results across three major open-weight families (Llama-2, Llama-3, and Qwen (Appendix G)). On the task side, as detailed in our response to W1, our experiments cover a wide range of domains, including creative, open-ended, and multilingual tasks. We believe this diverse evaluation provides strong evidence for the generality of our conclusions.
>
> > **Q1:** The manuscript would be strengthened by deeper engagement with related and contrasting work. Just to name a few, the recent paper by Rodemann et al. https://arxiv.org/abs/2502.14581 exploring implicit statistical biases via alignment deserve explicit discussion. Contra‐alignment perspectives such as the Overton Pluralism framework proposed by Lake et al. (https://arxiv.org/abs/2406.17692) are mentioned, but could be discussed in more detail. By omitting these debates, the paper risks overstating novelty and understating the broader scholarly context.
>
> **R1:** Thank you for these excellent recommendations. We agree that a more thorough discussion of related work is essential. The works you mentioned—Rodemann et al. on implicit statistical biases and Lake et al. on pluralism—are relevant. As you note, these works focus on the higher-level social and semantic consequences of current alignment practices, while our work provides a complementary, low-level analysis of the underlying probabilistic mechanism (i.e., probability concentration). They diagnose *what* happens at a societal level; we diagnose *how* it happens at a distributional level. In the revision, we have expanded Section 8 and 9 to detail this relationship, clarifying how our findings provide a foundational, quantitative underpinning for these important critical perspectives on alignment.
>
> > **Q2:** On the methodological side, the assumptions of ergodicity and stationarity required for AEP may be violated in highly variable prompts - open‐domain dialogue or code synthesis, for example - and quantitative error bounds under such conditions would bolster the argumetn.
>
> **R2:** This is a critical point, and we wish to clarify a key detail of our theoretical foundation. The version of the Asymptotic Equipartition Property (AEP) for LLMs that we use, following the work of Mudireddy et al. (2024), **does not require the assumptions of ergodicity or stationarity**, a point we state in Section 4.2.
>
> The proof (see Appendix H) relies only on the model being autoregressive and using finite-precision arithmetic—conditions that hold for virtually all modern LLMs. This makes our AEP-based estimator for BF robust and applicable even to the highly variable, non-stationary sequences generated from the diverse prompts used in our experiments, which include creative writing, news generation, and random strings.
>
> ### References
>
> [1] Dubey, A., Jauhri, A., Pandey, A., Kadian, A., Al-Dahle, A., Letman, A., ... & Ganapathy, R. (2024). The llama 3 herd of models. arXiv e-prints, arXiv-2407.

---

> > ### Comment · Reviewer_fLap · 2025-11-26
> >
> > Thank you for addressing my points so thoroughly and incoportating them in the revised version. I have decided to increase my score in light the changes made.

---

### Official Review · Reviewer_k2WN · 2025-11-03

**Soundness:** 3
**Presentation:** 3
**Contribution:** 2
**Rating:** 4
**Confidence:** 3

**Summary:**

The paper introduces a tool, called Branching Factor (BF), that can be used to study how LMs distribute probability over generations. For example, if we obtain multiple conditionally independent samples, given a prompt that is, and observe multiple, distinct strings, this situation corresponds to high BF. Conversely, if we only observe few, distinct strings, this situation corresponds to low BF.

The paper uses this tool to study the impact of model size and instruction tuning in the distributions over responses induced by different LMs. This kind of analysis can be informative when deciding on which model to use, or which decoding algorithm to use and by extension other things that affect decoding (e.g., prompt, few-shot examples, etc.).

The paper motivates the proposed tool using theory of stochastic processes, extended (by prior work) to LMs. It also motivates the tool intuitively by claiming that, unlike token level statistics, it is token-invariant. The tool was used to analyse a few models across a few tasks, supporting some insightful observations, but most practical implications of the use of this tool are left as discussion points in section 9.

**Strengths:**

1. The technique is simple and reasonably well-motivated
2. The paper is mostly clear (though I do find it to abuse of mathiness, which, in my reading, adds little)
3. The technique can power interesting decisions regarding decoding algorithms and/or models and/or prompting techniques.

**Weaknesses:**

I find the positioning of the work unclear and, as a result I perceive some mismatch between what it is claiming (or what it might be claiming) and what it delivers empirically.

Part of the motivation for this BS technique is 'token-invariance', but this, I believe, stayed at the level of argumentation only, with no empirical validation against techniques that do not deliver token-invariance. For example, would some 'not-token-invariant' technique lead to essentially different and/or misleading conclusions? (I am not claiming I know whether it will go one way or the other, it just looks like this should have been explored but it wasn't).

If I understand it correctly, BF is an exponentiated estimate of entropy (of the distribution over sequences, given a prompt), possibly aggregated across prompts depending on the analysis. Maybe the paper is claiming insight into analysing entropy (albeit exponentiated), or maybe it's claiming something else around the AEP result. Analysing entropy estimates does not come across as too surprising. On the other hand, the AEP result comes from prior work. So I am not too sure how to position this paper. Perhaps it is really about exploiting the AEP result in analysis, which would be fine as far as I am concerned, but somehow the presentation isn't clearly and transparently about 'just' that. The only issue then is that the AEP result is not contributing anything (if I understand it and its use here correctly) beyond 'licensing' us to interpret exponentiated entropy estimates (though entropy estimates are routinely interpreted, aren't they?).

**Questions:**

I would appreciate clarifications on the two points in weakness.

Also, I have some comments for clarity:

1. H(Y|x) is not conditional entropy, it's the entropy of a conditional rv. Conditional entropy would be taken in expectation under the joint distribution with x given random treatment, right?
2. I find the explanation in terms of perplexity rather confusing. Perplexity is a property of the model which we estimate on a data sample (like a dataset of human generated text). What you have is more like entropy, which you happen to exponentiate (for reasons that were clear without talking about perplexity), isn't it? And the point of connecting to an AEP result is to make a connection to a stochastic process' entropy rate (not perplexity), isn't it?

---

> ### Author Response · Authors · 2025-11-21
> **Responses to Reviewer k2WN [1/2]**
>
> We thank Reviewer k2WN for the thoughtful feedback. We are glad you found the technique simple, well-motivated, and capable of informing important decisions about model usage and decoding. We would like to clarify our positioning and the core contributions of our work.
>
> > **W1:** I find the positioning of the work unclear and, as a result I perceive some mismatch between what it is claiming (or what it might be claiming) and what it delivers empirically. Part of the motivation for this BF technique is 'token-invariance', but this, I believe, stayed at the level of argumentation only, with no empirical validation against techniques that do not deliver token-invariance. For example, would some 'not-token-invariant' technique lead to essentially different and/or misleading conclusions?
>
> **R1:** Thank you for this important question. Our primary research question is: "What is the effective Branching Factor for LLMs?" We aim to understand the concentration of the model's **entire output probability distribution**, a fundamental property. By "token-invariance," we mean that this distributional property should not be sensitive to surface-level factors like vocabulary size or the specific identities of high-frequency tokens, which can skew sample-based diversity metrics like Distinct-N.
>
> Our core contribution is not to propose BF as a novel metric to replace others, but to use it as a precise, theoretically-grounded lens to build a coherent, information-theoretic explanation for how alignment reshapes a model's generative possibilities. The empirical validation for this approach lies in the consistent patterns we find across different model families (Llama, Qwen) and a diverse set of tasks. The BF framework successfully connects and explains several disparate phenomena—such as the reduced sensitivity to decoding methods in aligned models and the stabilizing effect of CoT—which a purely sample-based, token-level metric would struggle to unify. We have expanded on the disconnect between BF and sample-based diversity in Appendix H, and we will sharpen this distinction in the main text.
>
> > **W2:** If I understand it correctly, BF is an exponentiated estimate of entropy (of the distribution over sequences, given a prompt), possibly aggregated across prompts depending on the analysis. Maybe the paper is claiming insight into analysing entropy (albeit exponentiated), or maybe it's claiming something else around the AEP result. Analysing entropy estimates does not come across as too surprising. On the other hand, the AEP result comes from prior work. So I am not too sure how to position this paper. Perhaps it is really about exploiting the AEP result in analysis, which would be fine as far as I am concerned, but somehow the presentation isn't clearly and transparently about 'just' that. The only issue then is that the AEP result is not contributing anything (if I understand it and its use here correctly) beyond 'licensing' us to interpret exponentiated entropy estimates (though entropy estimates are routinely interpreted, aren't they?).
>
> **R2:** We appreciate this question, which helps clarify our contribution. You are correct that BF is exponentiated entropy and that AEP is a known result. The novelty of our work lies not in the invention of a new metric, but in its application as a **unifying conceptual framework**. Our contribution is to show that the lens of probability concentration, as measured by BF, provides a powerful and unified explanation for a range of seemingly disconnected observations about aligned LLMs. Specifically, BF allows us to quantitatively connect:
>
> 1. Alignment tuning reduces BF → shrinks the generative horizon
> 2. Low BF explains decoding sensitivity → fewer viable options to prune
> 3. CoT stabilizes generation → shifts key information to late, low-BF regions
> 4. Avoid late branching → too many low-probability, low-quality continuations
> 5. Alignment surfaces low-entropy paths already latent in base models
>
> The AEP is a critical enabling tool that allows us to estimate BF efficiently for long sequences, but the core contribution is the explanatory framework built upon it. We will clarify this positioning more explicitly in the revised introduction.

---

> ### Author Response · Authors · 2025-11-21
> **Responses to Reviewer k2WN [2/2]**
>
> > **Q1:** H(Y|x) is not conditional entropy, it's the entropy of a conditional rv. Conditional entropy would be taken in expectation under the joint distribution with x given random treatment, right?
>
> **A1:** We apologize for the imprecise notation. $H(Y|x)$ in our work denotes the entropy of the conditional distribution $P(Y|X=x)$, conditioned on a specific instance $x$. It is not the conditional entropy $H(Y|X)$, which would involve an expectation over the distribution of $X$. We use this notation to emphasize that we are analyzing the output distribution for a *specific prompt*, which is central to our analysis. We have corrected and clarified this notation throughout the paper in the revision.
>
> > **Q2:** I find the explanation in terms of perplexity rather confusing. Perplexity is a property of the model which we estimate on a data sample (like a dataset of human generated text). What you have is more like entropy, which you happen to exponentiate (for reasons that were clear without talking about perplexity), isn't it? And the point of connecting to an AEP result is to make a connection to a stochastic process' entropy rate (not perplexity), isn't it?
>
> **A2:** This is an excellent point that highlights an important terminological distinction. As explained in General Response C3, there are two different notions of "perplexity":
>
> 1. **Perplexity of a probabilistic model** (NLP usage, [1]): This measures how well a model fits a reference dataset (e.g., human-generated text). It is computed as exponentiated cross-entropy on the reference data.
>
> 2. **Perplexity of a probability distribution** (information theory usage): This is defined as the exponentiated entropy of the distribution itself, which does not require a reference dataset.
>
> Our work uses the **second, information-theoretic definition**. We are measuring a property of the model's *own output distribution*, not its fit to external text. The AEP connects the entropy rate of this distribution to the log-likelihood of sequences sampled from it. We use the term "perplexity" in this context because it provides an intuitive interpretation of BF as the "effective number of choices" and connects to a long history of using perplexity to characterize distributions. However, we recognize the potential for confusion with the NLP-specific usage. In the revision, we introduce and distinguish these two definitions early on to ensure clarity in Introduction.
>
> ### References
>
> [1] Jurafsky, D., & Martin, J. H. (2025). Speech and Language Processing: An Introduction to Natural Language Processing, Computational Linguistics, and Speech Recognition with Language Models, 3rd edition.

---

### Official Review · Reviewer_BACX · 2025-11-03

**Soundness:** 1
**Presentation:** 1
**Contribution:** 3
**Rating:** 2
**Confidence:** 3

**Summary:**

This paper investigates the diversity of language model outputs—i.e., the effective “number” of possible continuations of $p(\mathbf{Y}\_{\geq t} \mid \mathbf{x} \circ \mathbf{y}\_{< t})$—for different prompts $\mathbf{x}$ and different timesteps $\mathbf{y}\_{< t}$.
To this end, the paper proposed branching factor, which (if I understand correctly) is the normalised per-token perplexity of the model’s full continuations:
$$B(\mathbf{x} \circ \mathbf{y}\_{< t}) = \exp\Bigg( E\_{\mathbf{y}\_{\geq t} \sim p(\mathbf{Y}\_{\geq t} \mid \mathbf{x} \circ \mathbf{y}\_{< t})} \bigg[ \frac{1}{|\mathbf{y}\_{\geq t}|} \sum\_{t’=1}^{|\mathbf{y}\_{\geq t}|} H(p(Y\_{t+t’} \mid \mathbf{x} \circ \mathbf{y}\_{< t + t’}))\bigg]\Bigg).$$

Note the entropy, in this equation, is taken for a single symbol at a time.
The paper than uses an Asymptotic Equipartition Property (AEP) for language models to estimate this value.
It shows that the branching factor decreases with $t$, meaning that later tokens have a smaller branching factor than earlier ones.
It also shows that instruction tuned models have an order of magnitude lower branching factor than “base” models.

**Strengths:**

Investigating how "concentrated" the probability distribution of language models is under different situations is an interesting research question.

The experiment "teacher-forcing" the base model's generation with a prefix generated by an instruction-tuned model is quite interesting.

**Weaknesses:**

Personally, I found this paper a bit hard to read and understand due to certain imprecisions.


**Definition of Branching Factor.** To define the `branching factor`, the paper first states there exists an “effective tree” $\mathcal{T}$ with high probability sequences $\mathbf{y}\_{\geq t}$. It, however, never defines exactly what it means by “high probability”. In section 4.1, the paper then re-defines $\mathcal{T}$ as the perplexity: $\exp\Big( H(p(Y\_{\geq t} \mid \mathbf{x} \circ \mathbf{y}\_{< t})) \Big)$, as this is the effective number of equally probable outcomes with the same total uncertainty. I appreciate this definition, and I think the authors could have started directly with this, instead of first informally introducing an “effective tree”, which in my opinion makes the definitions less precise and more confusing.

**Role of end-of-sequence; and fixing $N$ in definitions, but not in experiments.** The definition of branching factors relies on fixing the length of all strings to $N$. However, the experiments suggest an end-of-sequence token is used—as some plots seem to run longer than others. This creates an important mismatch between theory and experiments, in my opinion.

**Incorrect use of AEP for LLMs.** In equation 3, the authors write:
$$
 \lim\_{N \to \infty} P\Bigg( \bigg| -
\frac{1}{N} \log p(\mathbf{y}\_{\geq t} \mid \mathbf{x} \circ \mathbf{y}\_{< t}) -
H(p(\mathbf{Y}\_{\geq t} \mid \mathbf{x} \circ \mathbf{y}\_{< t}))
\bigg| < \epsilon \Bigg) = 0
$$
However, unless I’m mistaken, in the original cited paper (Mudireddy et al., 2024) the definition is:
$$
 \lim\_{N \to \infty} P\Bigg( \frac{1}{N} \bigg| -
\log p(\mathbf{y}\_{\geq t} \mid \mathbf{x} \circ \mathbf{y}\_{< t}) -
\sum_{t’=1}^{\infty} H(p(Y\_{t’} \mid \mathbf{x} \circ \mathbf{y}\_{< t + t’}))
\bigg| < \epsilon \Bigg) = 1
$$
With the conditional entropies being computed “locally”, one token at a time.
The AEP that the authors write down only holds in certain cases—most importantly, for ergodic processes.
Note also that this difference is smaller than epsilon with probability 1, meaning that it holds for *any* string $\mathbf{y}\_{\geq t}$ with non-zero probability mass.
Note as well that as the limit is taken with $N \to \infty$, these strings are infinite, meaning that:
$$
\frac{1}{N} \log p(\mathbf{y}\_{\geq t} \mid \mathbf{x} \circ \mathbf{y}\_{< t}) =
\frac{1}{N} \log p(\mathbf{y}\_{\geq t + 1} \mid \mathbf{x} \circ \mathbf{y}\_{< t + 1})
$$
As the log-probability of a single symbol cancels out in the infinite limit.
I believe this suggests that, if this AEP were correct, all plots should be constant (as appending an extra token $y\_{t + 1}$ to the conditioning prompt shouldn't change results).
In turn, I believe this suggests an issue in the theory.

**Questions:**

Could you expand on the role of end-of-sequence in the theory put forward by this paper, and in the experiments you ran? Is an end-of-sequence token used in the experiments? And, if yes, how do you choose an $N$ when estimating the branching factor?

Could you confirm that there is an issue with your use of the AEP? Or did I make any mistakes in my interpretation of your, or Mudireddy et al.’s (2024) work?

As I see it, the AEP results are not actually used in practice here: (i) you use end-of-sequence tokens, thus not simulating infinite-length in any way; (ii) you use 50 string-samples—instead of one very long sample—to estimate log-probabilities. Would it be fair to instead characterise your estimation procedure as using a simple monte carlo estimator?

---

> ### Author Response · Authors · 2025-11-21
> **Responses to Reviewer BACX [1/2]**
>
> We thank Reviewer BACX for the detailed and careful review. We appreciate the opportunity to clarify several important points regarding our methodology and theoretical grounding.
>
> > **W1: Definition of Branching Factor.** To define the *branching factor*, the paper first states there exists an "effective tree" *T* with high probability sequences *y_{1:L}*, however, never defines exactly what it means by "high probability". In section 4.1, the paper then re-defines *T* as the perplexity: *exp (H(p(y_t | x, y_{0:t-1})))* as this is the effective number of equally probable outcomes with the same total uncertainty. I appreciate this definition, and I think the authors could have started directly with this, instead of first informally introducing an "effective tree", which in my opinion makes the definitions less precise and more confusing.
>
> **R1:** We appreciate this feedback on clarity. In the revision, we have rewriten Section 4 to streamline the introduction of Branching Factor more clearly, grounding it immediately in exponentiated length-averaged entropy. The "effective tree" intuition is meant to provide conceptual scaffolding for why exponentiated entropy is a natural measure for the "effective number of choices".
>
> > **W2: Role of end-of-sequence; fixing *N* in definitions, but not in experiments.** The definition of branching factors relies on fixing the length of all strings to *N*. However, the experiments suggest an end-of-sequence is used—as some plots seem to come to run longer than others. This creates an important mismatch between theory and experiments, in my opinion.
>
> **R2:** Thank you for raising this important point. In our experiments we do generate until the model emits an end-of-sequence token, so individual samples can have different lengths. Our definition of BF already accommodates this: for each sampled sequence $y^{(i)}$ we compute the length-averaged entropy $\bar{H}(Y_{1:|y^{(i)}|} \mid x;\theta)$ (Note, it is $\bar{H}$, not $H$) using its realized length $|y^{(i)}|$, and the reported BF is the average of these per-sample quantities. This approach effectively **marginalizes over the empirical output length distribution** produced by the model for a given prompt, rather than fixing a single length $N$ in advance. This is a deliberate choice that allows us to analyze the impact of other factors (like alignment or model size) on BF without the confounding effect of a fixed, and potentially unnatural, generation length. We have made this explicit in the revision.
>
> > **W3: Incorrect use of AEP for LLMs.** In equation 3, the authors write: ... However, unless I'm mistaken, in the original cited paper (Mudireddy et al., 2024) the definition is: ...
>
> **R3:** We believe there is a misunderstanding here regarding our formulation of the Asymptotic Equipartition Property (AEP) for LLMs. Our use is correct and consistent with the cited work.
>
> In our paper, the AEP statement (Theorem 4.1, formerly Eq. 3) reads:
> $$ \lim_{N \rightarrow \infty}{P\left( \left\lvert -\frac{1}{N}\log \tilde{P}\left(y_{1:N} | x; \theta \right) - {\bar{H}\left(Y_{1:N} | x; \theta \right)}\right\rvert < \epsilon \right) } = 1 $$
>
> Please note two key points:
> 1. The right-hand side is **1**, not 0, because the theorem states that the probability of the deviation being smaller than $\epsilon$ converges to one.
> 2. The quantity inside the absolute value is expressed with the length-averaged entropy $\bar{H}(Y_{1:N} \mid x;\theta) = \frac{1}{N}\sum_{i=1}^N H(Y_i \mid [x, y_{1:i-1}];\theta)$, **exactly matching the AEP** in Mudireddy et al.’s Eq. (14), as reproduced below.
>
> We also believe the AEP definition you quoted from Mudireddy et al. may contain a small typo. Using
> their notations, they define $l_M(\mathbf{Y}\_N) \doteq -\frac{1}{N} \sum_{n=1}^N \log (p_n)$ (Eq. 3 in their paper, $p_n$ is the next token distribution, here **summation** over conditional log-prob gives the complete
> log-prob of the whole sequence) and $h_M(\mathbf{Y}\_N) \doteq \frac{1}{N}\sum_{n=1}
> ^N H(p_n)$ (Eq. 10, here summation over **conditional entropies** of each step gives
> **total entropy** up to $N$, not $\infty$), then, they state AEP (Eq. 14) as
> $$\lim_{N \rightarrow \infty}P{\left(|l_M(\mathbf{Y}_N) - h_M(\mathbf{Y}_N)| > \epsilon \right)} = 0$$
>
> This form (probability of a large deviation going to 0) is equivalent to our form (probability of a small deviation going to 1). In summary, our use of AEP for LLMs is correct and fully consistent with Mudireddy et al.

---

> ### Author Response · Authors · 2025-11-21
> **Responses to Reviewer BACX [2/2]**
>
> > **Q1:** Could you explain on the role of end-of-sequence in the theory put forward by this paper, and in the experiments you ran? Is an end-of-sequence token used in the experiments? And, if yes, how do you choose an *N* when estimating the branching factor?
>
> **A1:** See our response at W2.
>
>
> > **Q2:** Could you confirm that there is an issue with your use of the AEP? Or did I make any mistakes in my interpretation of your, or Mudireddy et al.'s (2024) work?
>
> **A2:** As explained in R3, we believe there is a misunderstanding. Our formulation is correct and consistent with the cited work.
>
> > **Q3:** As I see it, the AEP results are not actually used in practice here: (i) you use end-of-sequence tokens, thus not simulating infinite-length in any way; (ii) you use 50 string-samples—instead of one very long sample—to estimate log-probabilities. Would it be fair to instead characterise your estimation procedure as using a simple monte carlo estimator?
>
> **A3:** The practical estimator is indeed Monte Carlo, but it is *guided* by AEP. The theorem guarantees that once a sampled continuation is reasonably long, its length-averaged log-likelihood concentrates around the length-averaged entropy of the underlying distribution. We therefore draw $M$ samples (50 in our experiments), compute the per-sample length-averaged log-likelihoods, and average them.
>
> Here is the two-step logic:
>
> 1. **MC's Role (Marginalizing Length)**: While a standard Monte Carlo estimator might imply we are simply reducing variance for a fixed variable, our use of $M$ samples (50 in our experiments) serves a different, specific purpose: **marginalizing over the length distribution**. Since the model's output length is stochastic (determined by the EOS token) and unknown a priori, we cannot compute a closed-form expectation. We use Monte Carlo sampling to approximate the expectation over the natural distribution of sequences and their varying lengths as defined by the model. This marginalization helps avoid confounding effects for further analysis on impact factors for BF.
>
> 2. **AEP's Role (Justifying the Estimator)**: The AEP (as discussed in R3) guarantees that for a single, sufficiently long sequence of length $N$, its length-averaged log-likelihood ($-\frac{1}{N}\log p(y_{1:N})$) converges to the true length-averaged total entropy ($\bar{H}(Y_{1:N})$). This makes the length-averaged log-likelihood a valid and practical per-sequence estimator for the total  length-averaged entropy up to the same length $N$.
>
>
>
> ### References
>
> Mudireddy, A., Bell, T., & Mudumbai, R. (2024). Slaves to the law of large numbers: An asymptotic equipartition property for perplexity in generative language models. arXiv preprint arXiv:2405.13798.

---

> ### Comment · Reviewer_BACX · 2025-11-25
>
> Dear authors,
>
> > *W3: Incorrect use of AEP for LLMs.* In equation 3, the authors write: ... However, unless I'm mistaken, in the original cited paper (Mudireddy et al., 2024) the definition is: ...
>
> Let me address the response to this point first, since it seems like one of the most important issues to be resolved. It seems my point was not clear (and that this was not helped by the fact that I introduced three typos in these equations). I believe they should be:
>
> $$
>  \lim\_{N \to \infty} P\Bigg( \bigg| -
> \frac{1}{N} \log p(\mathbf{y}\_{\geq t} \mid \mathbf{x} \circ \mathbf{y}\_{< t}) -
> \bar{H}(p(\mathbf{Y}\_{\geq t} \mid \mathbf{x} \circ \mathbf{y}\_{< t}))
> \bigg| < \epsilon \Bigg) = 1
> $$
> and:
> $$
>  \lim\_{N \to \infty} P\Bigg( \frac{1}{N} \bigg| -
> \log p(\mathbf{y}\_{\geq t} \mid \mathbf{x} \circ \mathbf{y}\_{< t}) -
> \sum_{t’=1}^{N} H(p(Y\_{t’} \mid \mathbf{x} \circ \mathbf{y}\_{< t + t’}))
> \bigg| < \epsilon \Bigg) = 1
> $$
> *Hopefully*, there are no typos here anymore.
> But let me make the distinction here clearer.
> In the review, I said:
> > With the conditional entropies being computed “locally”, one token at a time.
>
> What I meant with this clause was, more precisely, that there is an important distinction between:
> $\frac{1}{N}\sum_{t’=1}^{N} H(p(Y\_{t’} \mid \mathbf{x} \circ \mathbf{y}\_{< t + t’}))$ and
> $\bar{H}(p(\mathbf{Y}\_{\geq t} \mid \mathbf{x} \circ \mathbf{y}\_{< t}))$. The authors state these things are the same, and it seems a bit awkward to try to correct them about their own definition, but I strongly believe this is not true. The distinction here is that the use of the random variable $\mathbf{Y}\_{\geq t}$ in the authors' definition implies the entropy is averaged across all possible $\mathbf{y}\_{\geq t}$ (as defined in line 150). However, in the AEP paper, $H(p(Y\_{t’} \mid \mathbf{x} \circ \mathbf{y}\_{< t + t’}))$ is computed **for a single string $\mathbf{y}\_{< t + t’}$**. This expectation makes a very important difference, and makes this paper's use of the AEP for LLMs incorrect. Mudireddy et al. (2025) say this explicitly after their eq. 11:
> > Note that the empirical entropy $h\_M(Y\_N )$ is not the same as $\frac{1}{N} H(P(Y\_N))$; the former expression is a random variable, whereas the latter is the entropy of [...] and thus is a constant number.
>
> I'll wait till we resolve this to answer the rest of the author responses, as this seems more important.
>
> Best,
> Reviewer

---

> ### Author Response · Authors · 2025-11-28
> **Follow-up Response**
>
> Dear Reviewer,
>
> We have carefully re-examined Mudireddy et al. (2024) (including their latest revision) and your derivation regarding W3. You are correct, and we sincerely apologize for the earlier confusion.
>
> We incorrectly conflated the limit of the negative log-likelihood (NLL) with the constant total entropy rate $\bar{H}(Y_{1:N})$. As you rightly pointed out, for non-ergodic processes like LLMs, the length-averaged NLL $-\frac{1}{N} \log p(\mathbf{y})$ converges (convergence in probability) to the *length-averaged realized entropies* $h_{\text{realized}}(y_{1:N}) \doteq \frac{1}{N}\sum_{t=1}^N H(Y_t | [x, y_{<t}]; \theta)$ (a random variable), rather than the expectation of that sum. Mudireddy et al. term this the "Un-Equipartition Property" to highlight that probability mass does not distribute uniformly across typical sets, but rather concentrates on paths with lower realized uncertainty.
>
> **Implications for our Estimator:** While our theoretical formulation of the limit in Theorem 4.1 was imprecise (we have fixed it in the revision), **the estimator used in our experiments remains valid.**
>
> Our goal is to estimate the Branching Factor, which relies on the total length-averaged entropy $\bar{H}(Y_{1:N})$.
> *(Note: For brevity below, we omit the dependence on prompt $x$ and parameters $\theta$, letting $p(\cdot)$ denote $p(\cdot | x; \theta)$.)*
>
> 1.  **Expectation of Realized Entropy is Total Entropy:** While individual sequences vary, the expectation of the realized entropy sum recovers the total entropy. By linearity of expectation and the chain rule of entropy:
>     $$
>     \begin{aligned}
>     \mathbb{E}\_{\mathbf{y}} \left[\frac{1}{N} \sum_{t=1}^N H(Y_t | Y_{<t} = \mathbf{y}\_{<t})\right] &= \frac{1}{N}\sum_{t=1}^N \mathbb{E}\_{\mathbf{y}} [H(Y_t | Y_{<t} = \mathbf{y}\_{<t})] \\\\
>     &=\frac{1}{N}\sum\_{t=1}^N p(\mathbf{y}\_{1:N}) H(Y_t | Y_{<t} = \mathbf{y}\_{<t}) \\\\
>     &= \frac{1}{N}\sum\_{t=1}^N \sum_{\mathbf{y}\_{<t}} p(\mathbf{y}\_{<t}) H(Y_t | Y_{<t} = \mathbf{y}\_{<t}) \underbrace{\sum_{\mathbf{y}\_{t:N}} p(\mathbf{y}\_{t:N} | \mathbf{y}\_{<t})}\_{=1} \\\\
>     &= \frac{1}{N} \sum\_{t=1}^N H(Y_t | Y\_{<t}) = \frac{1}{N} H(Y_{1:N}) = \bar{H}(Y_{1:N})
>     \end{aligned}
>     $$
>
> 2.  **NLL Approximates Realized Entropy:**
>     By the property you highlighted, for a single sampled sequence $\mathbf{y}$, the NLL approximates the realized entropy sum when $N$ is sufficiently large:
>     $$-\frac{1}{N} \log p(\mathbf{y}) \approx \frac{1}{N} \sum_{t=1}^N H(Y_t | Y_{<t} = \mathbf{y}_{<t})$$
>
> 3.  **Monte Carlo Estimator Validity:**
>     Therefore, by taking the Monte Carlo average of the NLL over $M$ samples (as we do in Eq. 3), we are effectively approximating the expectation of the realized entropies, which yields the target total entropy $\bar{H}(Y_{1:N})$.
>
> In fact, as demonstrated in Figure 2, the standard deviation of the NLL diminishes rapidly with length. This suggests that while the process is theoretically non-ergodic, the "effective" generation space shrinks such that the variance of the realized entropies is low, making the MC estimator highly efficient.
>
> **We have revised Section 4 and Theorem 4.1 to explicitly reflect this distinction, removing the incorrect AEP claim and replacing it with the "Un-Equipartition" logic that rigorously justifies using NLL as a proxy for realized entropy within an MC estimator.**
>
> Thank you for pushing us on this rigor; it has significantly clarified the theoretical grounding of the paper.

---

### Author Response · Authors · 2025-11-21
**General Response**

We thank the reviewers for their constructive feedback and for recognizing our work's novelty, technical soundness, strong experiment designs, and broad interests and impacts. We have revised the paper (major changes in blue) to address comments. Below we summarize our response to shared concerns and improvements.

### 1. Core Contribution Clarification
A key point of feedback (fwb1, k2WN, nL97) was that the novelty of the Branching Factor (BF) metric itself was limited, as it is closely related to distribution perplexity and average sequence entropy.
*   **Our Focus:** We investigate **probability concentration**—a fundamental distributional property—rather than just "output diversity." Two distributions can have similar diversity but vastly different structures (e.g., different high-probability paths).
*   **Unified Framework:** We refrain from proposing BF as merely "another diversity metric." Instead, we establish BF as a **unifying conceptual framework** that explains *why* aligned models behave as they do. This framework connects disparate observations:
    *   Alignment reduces BF $\rightarrow$ shrinks the generative horizon.
    *   Low BF $\rightarrow$ insensitivity to decoding strategies (Section 3).
    *   Chain-of-Thought $\rightarrow$ stabilizes generation by pushing decisive steps to late, low-BF regions (Section 6).
    *   Nudging $\rightarrow$ alignment surfaces latent low-entropy paths in base models (Section 7).

### 2. Theoretical Grounding: Theorem 4.1 (Response to BACX, k2WN, fLap)
We addressed concerns about the validity of Theorem 4.1 for LLMs:
*   **Theorem, not Assumption:** We clarify that Theorem 4.1 is a proven theorem (Mudireddy et al., 2024).
*   **Mild Assumptions:** It relies only on the model being **autoregressive** and using **finite-precision arithmetic**—conditions met by virtually all modern LLMs. It does *not* require ergodicity or stationarity.
*   **Estimator Validity:** We significantly rewrite Section 4 to explain how our estimator combines Monte Carlo sampling (for short sequences and length marginalization) and Theorem 4.1 (for long sequences) to provide robust estimates.

### 3. Clarifying "Perplexity" and BF Definition
We clarified BF's definition to avoid confusion (highlighted in the revised introduction section):
*   **Model Perplexity (NLP):** Measures fit to a reference dataset (exponentiated cross-entropy).
*   **Distribution Perplexity (Our Usage):** Measures the effective size of the model's *own* output distribution (exponentiated entropy).

### 4. Significant Changes in Revision
Beyond these clarifications, we have made substantive improvements based on reviewers' suggestions:
*   **Re-structured Formulation and Theory:** We rewrote Section 4 to streamline the introduction of BF more clearly and directly, and provide more details on practical computation (length marginalization, Monte-Carlo Estimator, and estimator for long sequences).
*   **New Section on Disentangling Alignment Stages:** A key limitation noted was that we did not disentangle which stage of alignment contributes most to BF reduction. We have added a **new paragraph in Section 7** with preliminary results on the OLMo2 model series, providing a more detailed analysis of how SFT and instruction tuning contribute to probability concentration.
*   **Expanded Related Works:** We have expanded Sections 8 and 9 to include the discussion for relevant work (Lake et al., 2024, Rodemann et al., 2025), clarifying how our findings provide a foundational, quantitative underpinning for these important critical perspectives on alignment, positioning our probabilistic work as the "how" complementing their "what." We also add a citation to Shi et al. (2024) in Section 3 and clarify the role of this section as a motivating example for the deeper analysis that follows.
*   **New Experimental Details:**
    *   **Robustness Verification (Table 2):** Added bootstrapping (100 runs) for low-temp sampling. Estimation variability is negligible ($\approx 0.01$), ensuring statistical significance.
    *   **Resampling:** Added detailed experiment setups and a clearer explanation of the resampling procedure (Section 6).
    *   **Nudging:** Clarified the use of small aligned models to "steer" large base models, isolating the effect of stylistic tokens (Section 7).
    *   **Model Coverage:** Highlight results for the **Qwen** model family and extension to multilingual tasks (Section 5.1)
*   **Societal Impact:** Added a discussion on the risks of "Homogeneity Bias" in alignment, connecting low BF to broader societal concerns.

We believe these comprehensive revisions have substantially improved the paper, clarifying its core contributions and strengthening its narrative. The result is a more focused and impactful paper that better highlights the value of the BF framework as a powerful diagnostic tool for understanding LLM outputs. We thank the reviewers again for their invaluable guidance in this process.

---

### Meta-Review · Area_Chair_Dofe · 2026-01-01

**Summary:**

This paper introduces Branching Factor as a metric to measure the probability concentration in LLM output distributions. Leveraging this metric, the authors examine the training dynamics of large language models and uncover several interesting and potentially important phenomena. These findings provide meaningful insights into LLM behavior and suggest a promising direction for better understanding model training and generation.

The current version still exhibits gaps between some theoretical claims and the supporting empirical evidence. Clarifying these connections and strengthening the experimental support would substantially improve the paper.  For these reasons, I do not believe the paper is yet ready for publication in its present form. Nevertheless, I am optimistic that, with revisions addressing these points, this work could develop into a strong and impactful contribution to the field.

**Reviewer Concerns:**

Addressed:
1) Stress-testing on evaluation and assumptions.
2) The novelty of the measurement.


Still outstanding:
1) This paper is a bit hard to read and understand.
2) The mismatch between what it is claiming (or what it might be claiming) and what it delivers empirically.
3) The motivation for the proposal of BF.

**Reviewer Scores:**

Reviewer BACX: The score may increase to 4 since part issues are solved.

Reviewer k2WN: The score would still be 4 since the core concern is not addressed.

Reviewer fLap: The score would increase to 6 since the concerns are addressed.

Reviewer nL97: The score would remain positive as 6.

Reviewer fwb1: The score would be kept as 2 since the core concern is not addressed.

---

### Decision · Program_Chairs · 2026-01-26

Reject